

# Large-scale transport into the Arctic: the roles of the midlatitude jet and the Hadley Cell

Huang Yang[1], Darryn W. Waugh[1,2], Clara Orbe[3], Guang Zeng[4], Olaf Morgenstern[4], Douglas
E. Kinnison[5], Jean-Francois Lamarque[5], Simone Tilmes[5], David A. Plummer[6], Patrick Jöckel[7], Susan
E. Strahan[8,9], Kane A. Stone[10,11,a], and Robyn Schofield[10,11]

[1]Department of Earth and Planetary Sciences, Johns Hopkins University, Baltimore, Maryland, USA
[2]School of Mathematics, University of New South Wales, Sydney, Australia
[3]NASA Goddard Institute for Space Studies, New York, New York, USA
[4]National Institute of Water and Atmospheric Research, Wellington, New Zealand
[5]National Center for Atmospheric Research (NCAR), Atmospheric Chemistry Observations and Modeling (ACOM)
Laboratory, Boulder, Colorado, USA
[6]Climate Research Branch, Environment and Climate Change Canada, Montreal, QC, Canada
[7]Deutsches Zentrum für Luft- und Raumfahrt (DLR), Institut für Physik der Atmosphäre, Oberpfaffenhofen, Germany
[8]Atmospheric Chemistry and Dynamics Laboratory, NASA Goddard Space Flight Center, Greenbelt, Maryland, USA
[9]Universities Space Research Association, Columbia, Maryland, USA
[10]School of Earth Sciences, University of Melbourne, Melbourne, Victoria 3010, Australia
[11]ARC Centre of Excellence for Climate System Science, University of New South Wales, Sydney, New South Wales 2052,
Australia
[a]now at: Department of Earth, Atmospheric and Planetary Sciences, Massachusetts Institute of Technology, Cambridge,
Massachusetts 02139-4307, USA

**Correspondence:** Huang Yang (hyang61@jhu.edu)

**Abstract.** Transport from the Northern Hemisphere (NH) midlatitudes to the Arctic plays a crucial role in determining the abundance of trace gases and aerosols that are important to Arctic climate via impacts on radiation and chemistry. Here we examine this transport using an idealized tracer with fixed lifetime and predominantly midlatitude land-based sources in models participating in the Chemistry Climate Model Initiative (CCMI). We show that there is a 20% - 40% difference in the Arctic

5 concentrations of this tracer among the models. This spread is found to be generally related to the spread in location of the Pacific jet, with lower Arctic tracer concentrations occurring in models with a more northern jet, during both winter and summer. However, the underlying mechanism for this relationship does not involve the jet directly, but instead involves differences in the surface meridional flow over the tracer source region, that vary with jet latitude. Specifically, in models with a more northern jet, the Hadley Cell (HC) generally extends further north and the tracer source region is mostly covered by

10 surface southward flow associated with the lower branch of the HC, resulting in less efficient transport poleward to the Arctic. During boreal summer, there are poleward biases in jet location in free-running models, and these models likely underestimate the rate of transport into the Arctic. Models using specified dynamics do not have biases in the jet location, but do have biases in the surface meridional flow, which results in differences in the transport into the Arctic. In addition to the land-based tracer, the midlatitude-to-Arctic transport is further examined by another idealized tracer with zonally uniform sources.

15 With equal sources from lands and oceans, the intermodel spread of this zonally uniform tracer is more related to variations





of parameterized convection over oceans than variations of HC extent particularly during boreal summer. This suggests that transport of land-based and oceanic tracers or aerosols towards the Arctic differ in pathways and therefore their corresponding intermodel variabilities result from different physical processes.

## 1   Introduction

The Arctic is characterized by the largest climate sensitivity with surface temperatures increasing much more rapidly than the global average in recent decades (IPCC, 2013). Trace gases and aerosols have been shown to be important for Arctic climate via their direct radiative influences and indirect effects on cloud properties (Garrett and Zhao, 2006; Lubin and Vogelmann, 2006; Coopman et al., 2018). Since the majority of these trace gases and aerosols originate over Northern-Hemisphere (NH) midlatitudes, where anthropogenic emissions are largest (Bottenheim et al., 2004; Fisher et al., 2010; Kupiszewski et al.,

2013), long-range transport from NH midlatitude source regions plays a crucial role in determining their Arctic distributions. Transport therefore has a remote impact on the Arctic climate as important as local forcings (Shindell, 2007). Shindell et al. (2008) further showed that the multi-model spread of simulated Arctic carbon monoxide (CO) and ozone ($O_3$) concentrations is as large as the corresponding multi-model mean and this large multi-model spread may be related to large differences in long-range transport. It is therefore important that models correctly represent this transport.

Orbe et al. (2018) recently analyzed the transport in models participating in the Chemistry Climate Model Initiative (CCMI), and showed a large spread among the models. A 30%-40% difference of the multi-model mean is found in the Arctic concentrations of idealized tracers originating in the NH midlatitudes. Orbe et al. (2018) attributed much of these differences in transport into the Arctic to differences in midlatitude (ocean) convective transport among the models, particularly during boreal winter. Specifically, they showed that, for tracers with zonally uniform sources over the northern midlatitudes, stronger convection over

the oceans tends to enhance tracer concentrations in the upper troposphere and dilute tracer concentrations at the surface. While this enhances transport into the upper troposphere, it also weakens along-isentropic transport from midlatitudes into the Arctic middle troposphere, manifesting as negative correlations between midlatitude convection and Arctic tracer concentrations.

A limitation of the study of Orbe et al. (2018) is that the authors focused on idealized tracers with zonally uniform sources, and hence it is unclear whether their conclusions apply to more realistic tracers. Most chemical tracers of interest have strong

zonal asymmetries in their source regions, with emissions primarily over midlatitude continents (e.g., tracers with anthropogenic emissions). These tracers may be less sensitive to differences in the simulated convection (which occur predominantly over the oceans), and there may be less spread among the models in the transport of these tracers into the Arctic.

Here we revisit the issue of transport into the Arctic within the CCMI models, considering an idealized "CO5O" tracer. This tracer has realistic, zonally-varying, emissions corresponding to anthropogenic carbon monoxide (CO) emissions but with

a idealized, fixed decay time of 50 days. We examine the transport of CO50, and also the "NH50" tracer (the same 50-day lifetime but with a zonally uniform boundary condition) considered by Orbe et al. (2018), into the Arctic within the CCMI models, and show there is a spread in Arctic concentrations of both tracers among the models. This spread of CO50 is, however,



not closely linked to differences in convective mass fluxes but rather to differences in the midlatitude jet over the Pacific Ocean and the mean meridional circulation.

Section 2 introduces the models, tracers, and dynamical metrics examined. Section 3 shows the multi-model mean tracer distribution followed by highlights on the multi-model spread of tracer concentrations over the Arctic. Section 4 focuses on examining the influences of jet on the poleward transport of tracers, in which we further explore the mechanisms. To further examine the role of tracer boundary condition and lifetime, Section 5 compares transport of CO50 (zonally asymmetric emissions) to the Arctic with another idealized tracer NH50 with zonally uniform sources as well as a realistic tracer carbon monoxide (CO) with a temporally and spatially varying chemical lifetime. Conclusions and discussions are given in Section 6.

## 2  Methods

### 2.1  Models and experiments

This study analyzes simulation results from models participating in the Chemistry Climate Model Initiative (CCMI) phase 1 (Morgenstern et al., 2017). CCMI is a joint activity of the international Global Atmospheric Chemistry (IGAC) and Stratosphere-troposphere Processes And their Role in Climate (SPARC) projects that aims to better quantify stratospheric and tropospheric ozone and other important chemical species using state-of-the-art chemistry-climate models (Eyring et al., 2013). Here, we examine distributions of the idealized CO50 and NH50 tracers (see Section 2.2) from 15 CCMs (Table 1). These models mostly overlap with those considered in Orbe et al. (2018), and we use the same model names. Several simulations analyzed by Orbe et al. (2018) are not used here because the CO50 tracer is not output in these simulations. As in Orbe et al. (2018), we focus on two types of hindcast reference simulations, namely the C1 simulation (i.e., referred to REF-C1 in CCMI) and the C1SD simulation (i.e., REF-C1SD in CCMI). Both C1 and C1SD simulations were forced by observed sea surface temperatures (SSTs) and sea ice concentrations (SICs) from the UK Met Office HadISST1 data set (Rayner, 2003), but they differ in the source of meteorological fields. C1 simulations calculate the meteorological fields within the model, whereas C1SD simulations use (or relax towards) meteorological fields from meteorological reanalyses. The models used in this study differ widely in many respects, including the model resolution, dynamic core, physical parameterization, and the chemical scheme (Table 1 and Morgenstern et al. (2017)).

We analyze monthly output from 2000 to 2009 for all CCMI simulations (except for GEOS-C1 and GEOS-C1SD, see Table 1), and calculate 10-year climatologies for northern winter and summer by averaging the months of December-January-February (DJF) and June-July-August (JJA), respectively. In addition, interpolation is applied from each simulation's native vertical levels (isobaric, hybrid pressure or hybrid altitude) to standard isobaric vertical coordinates consisting of 19 tropospheric levels (from 1000 hPa to 100 hPa with a uniform spacing of 50 hPa) and 4 stratospheric levels (at 80, 50, 30, and 10 hPa). Analysis of individual models is done using each model's native horizontal grid, but when forming multi-model mean fields, the model output is interpolated onto a standard $1° \times 1°$ grids at every isobaric level after the interpolation to common levels noted above.



## 2.2 Tracers

To quantify the large-scale transport from the NH midlatitude land sources to the middle-troposphere Arctic, we examine the idealized CO50 tracer. The CO50 tracer has a flux boundary condition, corresponding to the annual mean value of anthropogenic emissions of CO for 2000, from the Hemispheric Transport of Air Pollution (HTAP) REanalysis of the TROpospheric

chemical composition (RETRO) (Eyring et al., 2013), and a spatially uniform loss with a 50 day e-folding decay time. There are strong zonal asymmetries in the CO50 emissions (blue contours in Fig. 1), with the largest contributions from East and South Asia. Note that a similar 50 day CO-like tracer has also been used in many previous studies for diagnosing long-range transport, but in these previous studies the emissions included those from biomass burning (Shindell et al., 2008; Fang et al., 2011; Doherty et al., 2013, 2017).

We also compare the simulated CO50 field to the NH50 idealized tracer. The NH50 also has a spatially uniform 50-day loss, but with a different boundary condition. The concentration of NH50 ($\chi_{50}$) in the bottom model level is specified as a fixed mixing ratio (i.e., 10 ppmv) over the NH midlatitude region (30°N - 50°N, 180°E-180°W). Wu et al. (2018) have shown the spatial distribution of NH50, particularly its inter-hemispheric gradient, is strongly associated with the seasonal shift of the Intertropical Convergence Zone (ITCZ) and also likely the Southern Pacific Convergence Zone (SPCZ) over the oceans.

As noted in Section 1, Orbe et al. (2018) further documented a wide spread of NH50 concentrations amongst CCMI models both over the Arctic and in the SH, and attributed this spread to the intermodel variation of low-level parameterized convection primarily over the oceans.

Last, to examine how much CO50 can represent real tracers with land sources, we compare it with carbon monoxide (CO) that undergoes the full chemistry (spatially and temporally varying) in the models. CO is removed from the troposphere

primarily by reacting with the hydroxyl radical (OH) that yields a global mean annual mean lifetime of ∼2 months. However, as OH concentrations are much higher as well as for the temperature during summer, CO lifetime is much shorter in summer than that in winter. The emissions of CO generally resembles that of CO50, but it has additional sources from biomass burning, which features large emissions from forests in West Africa, South America all year round, as well as Siberia during summer. The latter is particularly important for the Arctic abundance of CO, and complicates comparisons with CO50.

## 25   2.3   Dynamical fields

As previous studies have indicated the importance of jet streams and storm track for tracer transport into the Arctic (e.g., Eckhardt et al., 2003), we also examine the relationship of the distributions of the above tracers with dynamical (meteorological) fields. In particular, we decompose the tracer transport into a zonally asymmetric component and a zonally symmetric component. The zonally asymmetric transport is associated with eddy mixing and we examine the relationship with the NH

midlatitude jet. The zonally symmetric transport is associated with the zonal-mean flow advection and we examine the relationship with the Hadley Cell (HC) circulation.

For the midlatitude jet, we focus on the zonal wind $u$ over the Pacific Ocean, as this plays an important role in the transport of CO50 away from the major source region over East Asia, and examine the variation of the latitude of the Pacific jet ($\phi_{\text{jet}}$). This





latitude is where the zonally (135°E-125°W) and vertically (500-800 hPa) averaged $u$ maximizes. To account for differences in model resolution, $\phi_{\text{jet}}$ is calculated as the location of the maximum of a quadratic fitted to the zonally and vertically averaged $u$ at its maximum grid point and the two points either side (Barnes and Polvani, 2013). $\phi_{\text{jet}}$ is calculated at every month of the integration and the wintertime and summertime climatologies of jet position are then derived for inter-model comparison.

For the HC we examine the 800 hPa - 950 hPa averaged zonal-mean meridional wind $\bar{v}$, and calculate the latitude ($\phi_{v=0}$) at which $\bar{v} = 0$ between 20°N and 50°N. This latitude corresponds to the surface divergence zone separating the NH Hadley and Ferrel Cells. Again, $\phi_{v=0}$ is calculated monthly, and winter and summer climatologies are compared between the models.

## 3    Distributions of CO50

We first examine the multi-model mean (i.e., C1 and C1SD simulations combined) distributions of CO50, and then examine
the spread amongst the models with a focus on distributions in the Arctic. The CCMI multi-model mean horizontal and vertical distributions of CO50 are shown in Fig. 1. In both the lower troposphere (850 hPa) and the middle troposphere (500 hPa), and for both seasons, there are higher concentrations of CO50 ($\chi_{\text{CO50}}$) over the midlatitudes than over the Arctic, with large zonal asymmetries over the midlatitudes but not the Arctic. The maxima of $\chi_{\text{CO50}}$ over the midlatitudes highlight the primary source regions of CO50 in the East Asia and the South Asia, with $\chi_{\text{CO50}}$ decreasing rapidly away from the source regions.

The meridional and vertical distribution of zonal-mean CO50 varies with season. During boreal winter, CO50 features much stronger meridional transport near the surface in both the poleward and equatorward directions. The distribution of CO50 also generally follows the slope of isentropic surfaces, with an enhanced vertical transport north of the midlatitude CO50 source region in contrast to a suppressed vertical transport in the south. During summer, $\chi_{\text{CO50}}$ has a weak vertical tracer gradient and a secondary maximum at 200 - 300 hPa in the subtropics, indicating reduced meridional transport compared to winter.
This secondary maximum in CO50 mixing ratio requires robust vertical transport with relatively slow chemical loss, which is likely due to close proximity of the emissions to the strong continental convection underlying the summertime Asian monsoon anticyclone over the Tibetan Plateau (Park et al., 2007; Garny and Randel, 2013) (see Fig. S1). A similar maximum within the Asian monsoon anticyclone region near the tropopause was observed by balloon sondes for particle surface area density of aerosols (Yu et al., 2017), as well as for CO in the upper troposphere over East Asia by flight measurements (Holloway et al.,
2000; Palmer et al., 2003).

The spatial distribution of zonal-mean CO50 for each model is similar to that for the multi-model mean distribution discussed above. This is illustrated in Fig. 2, which shows the latitudinal variation of lower troposphere (500 - 800 hPa) and vertical profiles of Arctic (70°N - 90°N) CO50, respectively. Although the latitudinal and vertical structure of variations in CO50 concentrations are similar among the models, there is a large spread in the magnitude of CO50 tracer concentrations. The
multi-model spread is the largest over the midlatitude source region, decreasing rapidly in the tropics south of the source region while remaining relatively large north of the source towards the Arctic for both seasons.

We will focus here on model differences in CO50 concentrations over the Arctic, and the poleward transport from NH midlatitudes. The differences in Arctic CO50 concentrations among the models peak around 400 hPa during both winter and



summer. In the middle and lower troposphere, the range of $\chi_{\mathrm{CO50}}$ among the models decreases from ∼11 ppbv in winter to ∼5 ppbv in summer. This yields a 20% - 40% wintertime fractional spread (i.e., multi-model spread relative to the corresponding multi-model mean) and a 25% - 30% summertime spread of Arctic $\chi_{\mathrm{CO50}}$.

The difference between pairs of simulations (and hence the ordering of simulations) is generally the same at all altitudes. For
example, $\chi_{\mathrm{CO50}}$ in winter is smaller in ACCESS-C1 and NIWA-C1 than that in the EMAC simulations throughout most of the tropospheric column. This suggests that the above model spread of Arctic CO50 is related to a vertically consistent difference in the poleward transport rather than a tracer redistribution between different levels.

The large spread in CO50 concentration among the models is consistent with the wide spread reported by Orbe et al. (2017, 2018) for idealized tracers with zonally-uniform sources. Also, Figs. 2 and S2 show that the spread in CO50 among the C1SD
simulations (dashed lines) is comparable to or even larger than the spread among the C1 simulations (solid lines). This is again consistent with the results of Orbe et al. (2017, 2018). This provides further evidence that using specified dynamics simulations does not constrain climatological tropospheric transport any more than using free-running models.

## 4    Transport processes of CO50

Having shown a large model spread in the Arctic concentrations of CO50, we now examine possible causes for these dif-
ferences. Shindell et al. (2008) suggested that the Arctic CO concentration in the middle troposphere is equally sensitive to changes of emissions over Europe, Asia, and North America. However, given the total amount of emissions from Asia (East Asia and South Asia, see Table 2 in Shindell et al. (2008)) is ∼2-3 times larger than those from Europe and North America, we first examine processes that are associated with the transport of Asian pollutants.

### 4.1    Relationship with midlatitude convection

Orbe et al. (2018) showed that differences in convection among models contribute to differences in tracer distributions. They showed that models with stronger midlatitude convection tend to have lower Arctic concentrations of the idealized NH5 tracer (this tracer has the same zonally symmetric boundary conditions as NH50 but a shorter lifetime of 5 days), especially during northern winter. However, examination for CO50 shows very weak relationship between the strength of the midlatitude convection and the Arctic $\chi_{\mathrm{CO50}}$ in both winter and summer (Fig. 3). The strength of convection is measured using the convective
mass flux (CMF) in the low-level midlatitudes, which is the average of 800 hPa - 950 hPa, 30°N-50°N, 130°E-170°E for boreal winter focusing on convection over West Pacific Ocean and 110°E-140°E for summer highlighting continental and maritime convection over East Asia, as in Orbe et al. (2018). The average zones overlap the strongest intensity and the largest intermodel variability of convection. Hence, variations in CMF do not seem to be the primary cause of variations in transporting CO50 into the Arctic. Note that the $\chi_{\mathrm{CO50}}$ - CMF relationship during winter is sensitive to models of choice as a positive correla-
tion would be established by excluding the ACCESS and NIWA models whereas a negative correlation by excluding EMAC simulation results. Such a sensitivity to inclusion or exclusion of a few models suggests that caution should be taken when interpreting the correlations from the limited number of CCMI models.





The absence of correlation between Arctic CO50 and midlatitude convection may be largely due to the zonally asymmetric boundary condition of CO50, particularly in winter. With primary sources over land, CO50 tends to be less impacted by the variability of convection that maximizes over the oceans during winter. In summer, despite midlatitude convection being the strongest and also having the largest model spread over land emissions, the poleward transport of CO50 along isentropic sur-

faces is much weaker than that in winter (comparing Fig. 1(f) with Fig. 1(e)). Therefore, the Arctic CO50 concentration during summer is less connected to CO50 concentration over the midlatitude sources and consequently shows a weaker correlation with the midlatitude convection.

## 4.2   Relationship with midlatitude jet

Figure 1(c,d) suggests that, in addition to convection, the zonal winds $u$, especially over the northern Pacific Ocean, also play an

important role in the transport of CO50 from its source region. We therefore start by examining the structure of the midlatitude jet over the Pacific Ocean in the models. Fig. 4 shows the latitudinal variation of lower-mid tropospheric (500 - 800 hPa) zonal wind $u$ averaged over the Pacific Ocean (135°E-125°W) for each model. In winter there is a similar latitudinal variation in $u$ among the models. There is a variation in the magnitude of the peak winds but the latitude of this peak $\phi_{\mathrm{jet}}$ varies by only a few degrees (∼35°N-40°N). The C1SD simulations, which use reanalyses winds, have very similar jet latitudes, with $\phi_{\mathrm{jet}} \sim$

36°N.

However, in summer, there is a much larger variation among the models, not only in the magnitude and location of peak winds but also the latitudinal structure. $\phi_{\mathrm{jet}}$ varies from ∼45°N to 57°N, with the C1SD models at the lower end $\phi_{\mathrm{jet}} \sim$ 45°N. This implies that the latitude of the Pacifc jet in C1 simulations is generally biased poleward of the reanalyses. A similar bias is found for models participating in the phase 5 of the Coupled Model Intercomparison Project (CMIP5) (Barnes and Simpson,

2017). The variation in the summertime jet structure among the models is further illustrated in Fig. 5, which shows the 500 hPa - 800 hPa averaged $u$ during summer in each individual simulation. The C1SD simulations show "strong" winds across the Pacific with axis tilting SE-NW. The C1 simulations show a much more varied structure, with many showing a more northern and more east-west jet that does not extend across the whole Pacific ocean.

The summertime distribution of the 500 - 800 hPa average CO50 concentration is also shown in Fig. 5 (colors), and there

appears to be a relationship between the midlatitude jet and the Arctic CO50 concentration. In general, lower $\chi_{\mathrm{CO50}}$ over the Arctic are found in simulations with a more northern jet over the Pacific Ocean and higher $\chi_{\mathrm{CO50}}$ in simulations with a more southern jet (primarily the C1SD simulations). The correspondence between the latitude of the Pacific jet and the Arctic CO50 during summer is further quantified in Fig. 6(a), which shows a scatter plot of Arctic $\chi_{\mathrm{CO50}}$ versus $\phi_{\mathrm{jet}}$ of the Pacific jet for summer. This shows that lower $\chi_{\mathrm{CO50}}$ is generally associated with a more northern $\phi_{\mathrm{jet}}$, with a clear negative correlation (-

0.83) between the climatological-mean values for each model. This suggests that models with a more northern jet have weaker (slower) midlatitude-to-Arctic transport.

Repeating the above analysis for winter, we find the wintertime tracer transport from NH midlatitudes to Arctic is also sensitive to jet location, with negative correlations between $\chi_{\mathrm{CO50}}$ and $\phi_{\mathrm{jet}}$, see Fig. 6(b). This is somewhat surprising given the differences in jet structure among models is much smaller in winter (see Fig. 4).





### 4.3 Mechanisms

We now explore the underlying mechanisms for the above connection between the Pacific jet and transport into the Arctic. A strong jet with rapid zonal flow at its center can act as a barrier to meridional transport (e.g., Bowman and Carrie, 2002), but there can also be intensive transport on the flanks of the jet due to Rossby wave breaking (RWB) (e.g., Haynes and Shuckburgh, 2000). This RWB on the edge of the jets may explain the connection between jet location and transport into the Arctic. As shown by the schematics in Fig. 7, when the Pacific jet is in a more northern position (e.g., summertime jets in C1 simulations as shown in Fig. 5) the source region of CO50 is on the equatorward flank of the jet and the anti-cyclonic RWB occurring here transports CO50 equatorward and blocks transport to the Arctic. In contrast, when the Pacific jet has a more southern position and its western end tilts more southward (e.g., summertime jets in C1SD simulations), a fraction of the CO50 source region overlaps the poleward flank of the jet and the cyclonic RWB occurring there transports CO50 to higher latitudes and the Arctic. In other words, differences in the Arctic $\chi_{\mathrm{CO50}}$ between models with different jet locations could be due to differences in the meridional eddy transport caused by RWB.

One approach to examine whether transport caused by RWB is the cause of differences in the transport into the Arctic is to decompose the tracer flux into a zonal-mean and an eddy components, i.e.,

$$\overline{v\chi_{\mathrm{CO50}}} = \overline{v}\,\overline{\chi_{\mathrm{CO50}}} + \overline{v'\chi'_{\mathrm{CO50}}}, \tag{1}$$

where $\overline{()}$ denotes the zonal mean, $()'$ is the corresponding departure from the zonal mean, $\overline{v\chi_{\mathrm{CO50}}}$ is the total flux, $\overline{v}\,\overline{\chi_{\mathrm{CO50}}}$ is the zonal-mean component, and $\overline{v'\chi'_{\mathrm{CO50}}}$ is the eddy component. The meridional fluxes are further vertically integrated to yield the corresponding tropospheric column flux across each latitude, i.e. the vertically integrated flux is

$$\langle F \rangle = \int\limits_{1000}^{p_0} F\, r_M\, dp\, a\cos\phi, \tag{2}$$

where $F$ is the total, mean, or eddy flux, $r_M$ is the ratio of molecular mass weight between CO (32 g/mol) and dry air (28.97 g/mol). $\phi$ is latitude, $p$ is pressure, $a$ is the Earth's radius of 6370 km, and $p_0$ is chosen as 200 hPa. Positive flux is defined as northward transport while negative corresponds to transport to the south.

A substantial contribution of the eddy flux comes from synoptic eddies, and to calculate this flux requires $v$ and $\chi_{\mathrm{CO50}}$ at higher frequency than the monthly-mean output available from the CCMI archive. However, we have access to daily output from GEOS-C1 and GEOS-C1SD simulations, which can be used to examine the relative roles of mean and eddy fluxes in the meridional transport. As the Arctic $\chi_{\mathrm{CO50}}$ in GEOS-C1 is much lower than that in GEOS-C1SD (with the difference being almost the largest among CCMI simulations in summer; see Fig. 6), comparison of the fluxes between these simulations can test whether differences in eddy transport are the causes of differences in Arctic CO50 concentrations.

The flux diagnostics for CO50 meridional transport in the two GEOS simulations are shown in Fig. 8. During summer there is an equatorward transport of CO50 in the subtropics and a poleward transport in the extratropics in both models (bold curves in Figs. 8(a,b)). The latitude separating the equatorward transport from the poleward transport shifts from $\sim$40°N in GEOS-C1 to $\sim$32°N in GEOS-C1SD. Given that CO50 is largely emitted from East Asia and South Asia over 20°N - 40°N, most of the



CO50 source region is characterized by equatorward transport in GEOS-C1 but a significant fraction of CO50 source stretches into the zone of poleward transport in GOES-C1SD. This yields a much larger poleward total flux over the midlatitudes in GOES-C1SD than that in GEOS-C1 (see Fig. 8(c)) consistent with a higher summertime Arctic $\chi_{CO50}$ in GEOS-C1SD than that in GEOS-C1. Examination of the zonal-mean and eddy fluxes shows that differences in the total fluxes are dominated by

the zonal-mean and not the eddy component (Fig. 8(c)). During winter, the latitude that separates the equatorward transport from poleward transport in GEOS-C1SD ($\sim 36°$N) is only slightly south of that in GEOS-C1 ($\sim 38°$N) (Fig. 8(d,e)). However, the total tracer flux of CO50 features a much larger poleward transport over the midlatitudes in GEOS-C1SD than that in GEOS-C1, and this large difference is again primarily due to difference in the zonal-mean fluxes.

The above analysis of tracer fluxes in GEOS-C1 and GEOS-C1SD contradicts our original speculation that the difference

in Arctic $\chi_{CO50}$ is due to the jet-associated RWB (and eddy transport). Instead, it indicates that differences in the zonal-mean component dominates, which is linked to transport by the meridional mean circulation. We are unable to perform this tracer flux decomposition in all CCMI simulations due to a lack of daily data. However, we can approximate the zonal-mean components of tracer flux using monthly-mean fields as the zonal-mean flux is largely associated with the slowly varying mean meridional circulation. We have confirmed that the zonal-mean flux calculated using monthly output differs only slightly from the one

using daily output in both GEOS-C1 and GOES-C1SD (see Fig. S3). The results for the approximated zonal-mean flux for each CCM are shown in Fig. 9. Note that the ACCESS-C1 and NIWA-C1 simulations are excluded for the analysis because $v$ in those simulations was output only at 850 hPa in the lower troposphere (800 hPa - 950 hPa) and cannot accurately represent the lower-tropospheric mean as compared to other CCMI simulations. After excluding the GEOS-C1 simulation in summer that shows opposite zonal-mean transport (i.e., equatorward flux north of source whereas poleward flux south of source, as shown

in Fig. 9(a)), the summertime spread of zonal-mean fluxes among CCMI models is comparable to that in winter ($\sim$100% of the multi-model mean). Moreover, in both seasons, the zonal-mean flux on the poleward flank of the midlatitude CO50 source region (latitudinal average of 40°N - 60°N and vertical average of 200 hPa - 1000 hPa) is generally positively correlated with the Arctic $\chi_{CO50}$, which again indicates that the difference in the zonal-mean flux is the major cause of the spread of Arctic CO50 concentrations among CCMI models.

There are also a few simulations deviating from this generally positive correlation between the zonal-mean flux and CO50 concentrations over the Arctic. For example, the zonal-mean flux in EMAC-L47-C1 is larger than that in EMAC-L47-C1SD, but the Arctic CO50 concentration is lower in EMAC-L47-C1. Analysis of 10-hourly output from EMAC simulations shows, consistent with the GEOS analysis above, that differences in zonal-mean fluxes is generally the major cause of differences in total flux between EMAC-L47-C1 and EMAC-L47- C1SD. However, this difference in flux is much smaller than that between

the two GEOS simulations, and does not explain the differences in the Arctic summer CO50 concentration between EMAC-L47-C1 and EMAC-L47-C1SD. Thus differences in other processes must be the causes for the differences between the two EMAC simulations. Further analysis is required to determine what this process is.

The above results suggest an important role of mean meridional circulations in the meridional transport of tracers, with larger poleward transport when the jet is located more equatorward. A possible reason for this connection between jet location and

transport by the mean meridional circulation could be the well-known link between the jet latitude and edge of the HC (e.g.,



Staten and Reichler, 2014) (also noted in Fig. 7). Specifically, it is shown that when the midlatitude jet is located more poleward, the HC extends further poleward. Thus, when the Pacific jet is in a more northern position, as in most of C1 simulations, the HC likely also extends further north and the CO50 source region is mostly covered by the lower branch of the HC with southward surface flow, and this may result in less poleward transport. Fig. 10(a) shows the meridional profile of summertime low-level

(800 hPa - 950 hPa averaged) zonal mean meridional wind $\bar{v}$. While there is an agreement in the general shape of the latitudinal variation of $\bar{v}$, there is a large spread in the magnitude of the flow and, most importantly, in the latitude where the flow changes from northerly to southerly.

To examine this possible relationship we use $\phi_{v=0}$ (see details in Section 2.3) to identify the latitude where the surface meridional flow $v$ changes from southward to northward flow. $\phi_{v=0}$ varies from 32°N - 45°N among the models, with even

a spread of 32°N to 38°N for C1SD simulations. Furthermore, there is a negative correlation between $\phi_{v=0}$ and the Arctic $\chi_{CO50}$ (Fig. 10(b)), i.e., when $\phi_{v=0}$ is further north (south), there is less (more) poleward transport of CO50. The spread in $\bar{v}$ and $\phi_{v=0}$ among the models is smaller in winter, but there is again a negative correlation between $\phi_{v=0}$ and the Arctic $\chi_{CO50}$, see Fig. 10(d-e).

As noted above, previous studies have shown a connection between the latitudinal extent of the HC and the latitude of the

midlatitude jet. This is also the case for $\phi_{v=0}$ and $\phi_{jet}$, with a positive correlation between these quantities, see Fig. 10(c,f). This explains why a negative correlation is also found between $\phi_{jet}$ and $\chi_{CO50}$. Close inspection of Fig. 10(c,f) shows that there is a tight $\phi_{jet}$ - $\phi_{v=0}$ relationship for the C1 simulations, but a large spread for C1SD simulations. The C1SD simulations agree in latitude of the jet but there is a large spread (comparable to spread amongst C1 simulations) in $\phi_{v=0}$ (and a corresponding spread in $\chi_{CO50}$) despite both $u$ and $v$ being constrained by reanalyses in C1SD simulations. In fact, $\phi_{v=0}$ in C1SD simulations

are generally biased equatorward comparing with those in reanalyses, which partially contributes to the higher Arctic $\chi_{CO50}$ in C1SD simulations especially during summer.

In summary, we have proposed two mechanisms, illustrated in Fig. 7, for why there are generally larger Arctic CO50 concentrations in models with a more southern location of the Pacific jet: The first mechanism relates directly to a shift in jet location and associated changes in Rossby wave breaking, while the other mechanism does not involve the jet directly but

instead relates to the surface meridional flow that varies consistently with the jet. Analysis of the zonal-mean and eddy tracer fluxes indicates that the second mechanism is the dominant cause of the spread in Arctic CO50 concentrations among the models, i.e. differences in the mean meridional circulations among models and the direction of surface meridional flow over the CO50 source region are the keys for the spread in poleward transport among the models.

## 5  Comparison with other tracers

The above analysis has suggested that variations in the near-surface extent of the HC (latitude where $v = 0$) among the models is a major contributor to the spread in transport of CO5O to the Arctic, and that variations in CMF play a minor role. This appears to contradict the studies of Orbe et al. (2017, 2018), which show that variations in CMF play a large role in the spread of the tracers they considered (i.e., NH5 as noted in Section 4.1). We also expect an important role of HC extent and associated





zonal-mean transport for Arctic NH50, since NH50 has a zonally uniform boundary condition. We therefore revisit the spread in NH50 among the CCMI models, to compare the relative roles of CMF and HC extent. We also, briefly, examine CCMI model simulations of CO to see if there is also an impact on realistic tracers with full chemistry.

### 5.1 NH50

The multi-model mean distribution of NH50 features a stronger transport along isentropic surfaces so that Arctic $\chi_{\mathrm{NH50}}$ peaks in the middle troposphere ($\sim$400 hPa, see gray lines in Fig. 11(c,d) and also in Fig. S4). Similar to CO50, the spread of NH50 concentrations among models is the largest in midlatitudes and remains almost unchanged further north. The spread in the Arctic concentrations of NH50, in particular, is also comparable to that for CO50 (see Fig. 11; latitude and verical profiles of each model), with a fractional spread of 20 - 25% in winter and 40 - 50% in summer. The overall similarity between CO50

and NH50 is further indicated in Fig. 12(a,e) with moderate correlations between the Arctic concentrations of the two tracers (0.58 in summer and 0.42 in winter). Again. the $\chi_{\mathrm{CO50}}$ - $\chi_{\mathrm{NH50}}$ correlation during winter is sensitive to models of choice. The positive correlation presented in Fig. 12(e) is largely due to ACCESS and NIWA, and oppositely a negative correlation is rendered if these two models are excluded.

  To explore the relative role of changes in CMF, latitude of the Pacific jet ($\phi_{\mathrm{jet}}$), and HC extent ($\phi_{v=0}$) in causing the spread in

NH50, we repeat the above analyses in Section 3 and examine the correlations of Arctic concentrations of NH50 with different quantities, see Fig. 12. In contrast to CO50, there is a stronger relationship of NH50 with CMF, especially during winter, see Fig. 12(f). Consistent with the study of Orbe et al. (2018) for NH5, Arctic NH50 concentrations tend to be lower in simulations that feature larger low-level mid-latitude CMF and such a correlation is weaker in summer. The summertime CMF - NH50 correlation (-0.09) is much lower than the CMF - NH5 correlation (-0.45) reported by Orbe et al. (2018). This is not due to a

difference between NH50 and NH5 but rather different models used in the two studies. GEOS-CTM is included here but not by Orbe et al. (2018), and it has higher NH50 than models with similar CMF (and lowers the correlation). At the same time MRI simulations are included by Orbe et al. (2018), but not here (as they do not include the CO50 tracer). These MRI results have high CMF and low NH5 and thus increase the correlation.

  Unlike CO50, NH50 exhibits only a moderate or weak correlation with $\phi_{\mathrm{jet}}$ and $\phi_{v=0}$. Note that the $\phi_{\mathrm{jet}}$ - NH50 seems

to be stronger during winter (-0.5), but this correlation is largely due to the ACCESS and NIWA results. Without these two models, a moderate positive correlation is found instead, which is consistent with Fig. 12(g) showing a moderate $\phi_{v=0}$ - NH50 correlation during winter.

  In summary, despite NH50 having a zonally uniform boundary condition, the multi-model spread of Arctic NH50 seems to be much less impacted by differences in the HC extent and associated zonal-mean transport among models. Instead, NH50

shows a stronger correlation with low-level mid-latitude convection especially during boreal winter, as shown by Orbe et al. (2018). Therefore, in contrast to a minor role for transporting CO50 towards the Arctic, midlatitude convection predominantly contributes to the intermodel variations of Arctic NH50 concentrations in winter. In summer, convection may play a role as important as the HC extent (as for CO50), but the two processes may act oppositely so that correlations of summertime NH50



are weaker for both. The above results again suggest that transport of zonally uniform (or oceanic) tracers differ in pathways compared to land tracers, and low-level convection over the oceans seem to play a more significant role.

## 5.2 CO

It is also of interest to examine whether the above conclusions based on idealized tracers apply to more realistic tracers with
interactive chemistry. We therefore examine whether the spread, and relationship with the HC extent (i.e. $\phi_{v=0}$), found in CO50 can also be found in full chemistry simulations of CO from the CCMI models.

The comparison of CO50 and CO from CCMI results shows a positive relationship between the Arctic concentration of CO50 and CO in winter, but no relationship in summer; see Fig. 13(a,c). This suggests that differences in transport that cause differences in CO50 might explain a significant fraction of the multi-model spread of CO during winter when chemistry is
relatively weak, but these transport differences are likely less important during summer when model differences in chemistry dominate. This is borne out in Fig. 13(b,d), which shows a weak-moderate negative relationship between $\phi_{v=0}$ and $\chi_{CO50}$ during winter but no relationship during summer. This indicates that chemistry may still determine the spatial distribution of real tracers, especially during summer when tracers are more chemically reactive. As to variations of chemistry among models, a detailed examination on the spatiotemporal variability of tropospheric OH is needed.
In addition to chemistry, differences in emissions between CO and CO50 are also likely to result in their differed sensitivities to variations of the HC extent among the models. In particular, CO features an additional summertime emission from biomass burning over Siberia, which is in close proximity to the Arctic and hence tends to have a strong influence on the Arctic CO concentration. However, this emission region is distant from the HC edge over the NH midlatitudes and tracer transport from this region is less likely to be impacted by variations of the HC extent.

## 6   Conclusions

In this study, we examine long-range transport into the Arctic by an idealized CO5O tracer with predominantly midlatitude Asian emissions in simulations from a suite of CCMs. There is a wide spread (20 - 40%) of the Arctic concentrations of CO50 among the simulations, indicating a large intermodel variability in the simulated NH midlatitude-to-Arctic transport. Further, this spread is found to be related to the location of the Pacific jet among the models, with lower Arctic tracer concentrations
for a more northern Pacific jet. While the spread in transport to the Arctic is associated with the latitude of the jet, our analysis indicates that this is an indirect relationship, with difference in the mean meridional flow (that is correlated with the jet latitude) being the dominate cause of differences in the poleward transport of tracers. Specifically, in models with a more northern jet, the Hadley Cell (HC) generally extends further north and the tracers source region is mostly covered by the lower branch of the HC with southward surface flow, and resulting in less poleward transport. Differences in midlatitude convection among the
models appear to play a secondary role.

While the spread in Arctic CO50 concentrations is determined primarily by the HC extent, this is not the case for the NH50 tracer that features zonally uniform midlatitude sources, which shows a larger correlation with midlatitude convection



over the Pacific Ocean during winter, as shown for NH5 by Orbe et al. (2018). Thus, it is likely that variations in convection over the oceans are more efficient in influencing the transport of trace species towards the Arctic than variations in surface meridional flow during winter. Specifically, for NH50 that has similar sources from oceans and lands, the role of convection over the oceans overweights the influences of surface meridional flow. In contrast, for CO50 which has emissions primarily over

land, variations in convection over the oceans are remote and less influential and therefore zonal-mean transport by surface meridional flow dominates. In summer, the relative importance of convection versus HC extent is more complex for NH50 suggesting comparable and offsetting effects from both convection and surface meridional flow.

The free-running model C1 simulations have a jet on average further poleward than observed during summer (a common bias in climate models (Barnes and Simpson, 2017)), with a corresponding bias in the latitudinal extent of the HC. The corre-

lation between the transport into the Arctic and the latitude of the jet (or the HC edge) then implies that these models likely underestimate the transport into the Arctic. While we have focused on impacts on the CO50 and NH50 tracers, this bias likely exists for the transport of other tracers with predominantly land sources and relatively long lifetimes. Therefore, free-running climate models may underestimate the rate of transport into the Arctic for radiatively important (long-lived) gases, especially during summer.

The specified dynamics simulations (C1SD) which use the same (or very similar) specified meteorological fields do not have bias in jet location, but, surprisingly, there is a spread in the latitude where $v = 0$ (i.e., $\phi_{v=0}$), which results in a spread in the rate of transport into the Arctic. Orbe et al. (2017, 2018) also noted a spread in the transport among C1SD simulations, which they related to the spread in transport due to differences in the parameterized convective mass fluxes. Here we have shown that variations in near-surface $v$ is also a major contributor to differences in transport among C1SD models. Analysis of other

metrics of the HC extent show agreement among C1SD simulations for a metric based on $u$ (latitude where surface zonal wind vanishes) but a larger spread for a metric based on $v$ (latitude where mean meridional stream function at 500 hPa switches the sign) [person. comm. with Orbe]. It is an open question as to why the C1SD simulations agree on the latitude of the Pacific jet, but not on the latitude where $v = 0$. One possible explanation is that the unconstrained bias of $u$ in C1SD simulations are similar to those of $v$ (similar results are shown by Uhe and Thatcher (2015) in the ACCESS model). Given that $u$ is much

larger than $v$ in magnitude, bias of $u$ is likely to be much smaller than its absolute magnitude and hence results in negligible bias in the location of the Pacific jet while bias of $v$ can be comparable to its absolute magnitude causing a large spread in the determined HC extent. Further analysis is required of not only the convective mass flux but also the surface meridional flow for illustrating variations in the NH midlatitude-to-Arctic transport driven by dynamical processes, particularly those variations in C1SD simulations.

*Data availability.* Most of data from CCMI-1 used in this study can be obtained through the British Atmosphere Data Centre (BADC) archive (ftp://ftp.ceda.ac.uk, last access: 06 July 2018). Data from Community Earth System Model (CESM) can be obtained through the Climate Data Gateway at National Center for Atmospheric Research (NCAR) (https://www.earthsystemgrid.org/search.html?Project=CCMI1, last



access: 6 July 2018). For instructions for access to both archives, see http://blogs.reading.ac.uk/ccmi/badc-data-access/ (last access: 2 August 2018)

*Competing interests.*  no competing interests

*Acknowledgements.*  We thank the Centre for Environmental Data Analysis (CEDA) for hosting the CCMI data archive. We acknowledge the modeling groups for making their simulations available for this analysis and the joint WCRP SPARC/IGAC Chemistry–Climate Model Initiative (CCMI) for organizing and coordinating this model data analysis activity. In addition, Clara Orbe want to acknowledge the high-performance computing resources provided by the NASA Center for Climate Simulation (NCCS) and support from the NASA Modeling, Analysis and Prediction (MAP) program. Olaf Morgenstern and Guang Zeng acknowledge the UK Met Office for use of the MetUM. Their research was supported by the NZ government's Strategic Science Investment Fund (SSIF) through the NIWA program CACV. Olaf Morgenstern acknowledges funding by the New Zealand Royal Society Marsden Fund (grant 12-NIW-006) and by the Deep South National Science Challenge (http://www.deepsouthchallenge.co.nz). Olaf Morgenstern and Guang Zeng also wish to acknowledge the contribution of the NeSI high-performance computing facilities to the results of this research. New Zealand's national facilities are provided by the New Zealand eScience Infrastructure (NeSI) and funded jointly by NeSI's collaborator institutions and through the Ministry of Business, Innovation and Employment's Research Infrastructure program (https://www.nesi.org.nz). Huang Yang and Darryn W. Waugh acknowledge support from NSF grant AGS-1403676. Darryn W. Waugh acknowledges NASA grant NNX14AP58G. Huang Yang thanks discussions with Dr. Luke D. Oman, and helps from Drs. Andreas Pfeiffer and Sabine Brinkop for accessing 10-hourly EMAC outputs. The EMAC model simulations have been performed at the German Climate Computing Centre (DKRZ) through support from the Bundesministerium für Bildung und Forschung (BMBF). DKRZ and its scientific steering committee are gratefully acknowledged for providing the HPC and data archiving resources for the consortial project ESCiMo (Earth System Chemistry integrated Modeling). Robyn Schofield and Kane A. Stone acknowledge support from the Australian Research Council's Centre of Excellence for Climate System Science (CE110001028), the Australian government's National Computational Merit Allocation Scheme (q90) and the Australian Antarctic science grant program (FoRCES 4012).



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



**Table 1.** Selected period, horizontal and vertical configurations of simulations. FD is finite difference; FV is finite volume; STL is spectral transform linear; STQ is spectral transform quadratic; TA is hybrid terrain-following altitude; TP is hybrid terrain-following pressure; P is pressure.

| Simulations | Selected period | Hor. resolution (lat×long) | Hor. discretization | Vert. levels | Top level | Coord. sys. |
|---|---|---|---|---|---|---|
| ACCESS-C1 | 01/2000-12/2009 | $2.5° \times 3.75°$ | FD | 60 | 84 km | TA/P[b] |
| CMAM-C1 | 01/2000-12/2009 | $\sim 3.8° \times 3.8°$ (T47) | STL | 71 | 0.08 Pa | TP |
| CMAM-C1SD | 01/2000-12/2009 | $\sim 3.8° \times 3.8°$ (T47) | STL | 71 | 0.08 Pa | TP |
| EMAC-L47-C1 | 01/2000-12/2009 | $\sim 2.8° \times 2.8°$ (T42) | STQ | 47 | 1 Pa | TP |
| EMAC-L47-C1SD | 01/2000-12/2009 | $\sim 2.8° \times 2.8°$ (T42) | STQ | 47 | 1 Pa | TP |
| EMAC-L90-C1 | 01/2000-12/2009 | $\sim 2.8° \times 2.8°$ (T42) | STQ | 90 | 1 Pa | TP |
| EMAC-L90-C1SD | 01/2000-12/2009 | $\sim 2.8° \times 2.8°$ (T42) | STQ | 90 | 1 Pa | TP |
| GEOS-C1 | 01/1990-12/1998[a] | $2° \times 2°$ | FV | 72 | 1.5 Pa | TP |
| GEOS-CTM | 01/2000-12/2009 | $2° \times 2.5°$ | FV | 72 | 1.5 Pa | TP |
| GEOS-C1SD | 01/2000-12/2007[a] | $2° \times 2°$ | FV | 72 | 1.5 Pa | TP |
| WACCM-C1 | 01/2000-12/2009 | $\sim 1.9° \times 2.5°$ | FV | 66 | 140 km | TP |
| WACCM-C1SDV1 | 01/2000-12/2009 | $\sim 1.9° \times 2.5°$ | FV | 88 | 140 km | TP |
| WACCM-C1SDV2 | 01/2000-12/2009 | $\sim 1.9° \times 2.5°$ | FV | 88 | 140 km | TP |
| CAM-C1 | 01/2000-12/2009 | $\sim 1.9° \times 2.5°$ | FV | 26 | 200 Pa | TP |
| CAM-C1SD | 01/2000-12/2009 | $\sim 1.9° \times 2.5°$ | FV | 56 | 200 Pa | TP |
| NIWA-C1 | 01/2000-12/2009 | $2.5° \times 3.75°$ | FD | 60 | 84 km | TA/P[b] |

[a] Negligible differences in climatology between these simulations and another two corresponding GEOS simulations averaged with the period of 01/2000-12/2009.

[b] Simulations are based on the TA coordinates, but the meteorological fields have been particularly interpolated into the P coordinates with 31 levels.





**Table 2.** Tracers and dynamical/thermodynamic variables of models analyzed in the study. $u$ and $v$ are the zonal wind and meridional wind; CMF is the Convective Mass Flux by moist convection updraft. Available variables are marked by "x". Variables of some simulations are scaled before the inter-model comparison and details are listed in table footnotes. (Re)analysis for specified dynamics in C1SD simulations are listed in the second last column, otherwise the meteorology is free-running (FR) in C1 simulations. Moreover, meteorological fields are specified by nudging in most of C1SD simulations, except GEOS-CTM uses CTM and GEOS-C1SD uses replay. The moist convection scheme is listed in the last column.

| Simulations | CO50 | NH50 | CO | $u$ | $v$ | CMF | Meteor. fields | Moist conv. schemes |
|---|---|---|---|---|---|---|---|---|
| ACCESS-C1 | x | x | x | x | x[a] | x[b] | FR | Hewitt et al. (2011) |
| CMAM-C1 | x | | x | x | x | x | FR | Zhang and McFarlane (1995) |
| CMAM-C1SD | x | | x | x | x | x | ERA-interim | Zhang and McFarlane (1995) |
| EMAC-L47-C1 | x | x[c] | x | x | x | x | FR | Tiedtke (1989); Nordeng (1994) |
| EMAC-L47-C1SD | x | x[c] | x | x | x | x | ERA-interim | Tiedtke (1989); Nordeng (1994) |
| EMAC-L90-C1SD | x | x[c] | x | x | x | x | ERA-interim | Tiedtke (1989); Nordeng (1994) |
| GEOS-C1 | x | x[d] | | x | x | x | FR | Moorthi and Suarez (1992); Bacmeister et al. (2006) |
| GEOS-CTM | x | x | | x | x | x | MERRA | Moorthi and Suarez (1992); Bacmeister et al. (2006) |
| GEOS-C1SD | x | x[d] | | x | x | x | MERRA | Moorthi and Suarez (1992); Bacmeister et al. (2006) |
| WACCM-C1 | x | x | x | x | x | x | FR | Hack (1994); Zhang and McFarlane (1995) |
| WACCM-C1SDV1 | x | x | x | x | x | x | MERRA[e] | Hack (1994); Zhang and McFarlane (1995) |
| WACCM-C1SDV2 | x | x | x | x | x | x | MERRA[f] | Hack (1994); Zhang and McFarlane (1995) |
| CAM-C1 | x | x | x | x | x | x | FR | Hack (1994); Zhang and McFarlane (1995) |
| CAM-C1SD | x | x | x | x | x | x | MERRA[e] | Hack (1994); Zhang and McFarlane (1995)- |
| NIWA-C1 | x | x | x | x | x[a] | x[b] | FR | Hewitt et al. (2011) |

[a] There are only two levels below 800 hPa (i.e., 850 hPa and 1000 hPa) for model output in P coordinates. Therefore, further vertical interpolation is problematic near the surface due to large impacts of topography at 1000 hPa, and analyses on lower-troposphere $v$ and related diagnosis of the Hadley Cell exclude these simulations.

[b] Scaled by 1/9.80665.

[c] Scaled by 100.

[d] Scaled by 0.001.

[e] Relaxing timescale is 50 hours for nudging.

[f] Relaxing timescale is 5 hours for nudging.



**Figure 1.** Ensemble mean of the horizontal distribution of CO50 concentration (shades, units: ppbv) at levels of (a,b) 850 hPa and (c,d) 500 hPa, and ensemble mean of the meridional and vertical transport of CO50 by showing the zonal-mean cross sections (e,f) during (a,c,e) DJF and (b,d,f) JJA. In (a,b,c,d), horizontal winds $(u,v)$ are overlaid as vectors and regions of CO50 sources are highlighted by white stipples within which CO50 emission fluxes are larger than $0.4 \times 10^{-9}\,\mathrm{kg/m^2/s}$. In (e,f), isentropic surfaces are overlaid as dark gray isopleths (units: K), the tropopause is marked as the bold dark gray curve, and regional convective mass flux (CMF) over the East Asia (110°E-140°E) are denoted by white contours (units: $10^{-3}\,\mathrm{kg/m^2/s}$).





**Figure 2.** (a,b) Latitudinal and (c,d) vertical variations of zonal mean CO50 concentration $\chi_{CO50}$ (units: ppbv) for each model during (a,c) DJF and (b,d) JJA. The corresponding ensemble means are depicted as the heavy gray lines. In (a,b), $\chi_{CO50}$ are averaged in the middle and lower troposphere (500-800 hPa); whereas in (c,d), $\chi_{CO50}$ are averaged over the Arctic (70°N-90°N).





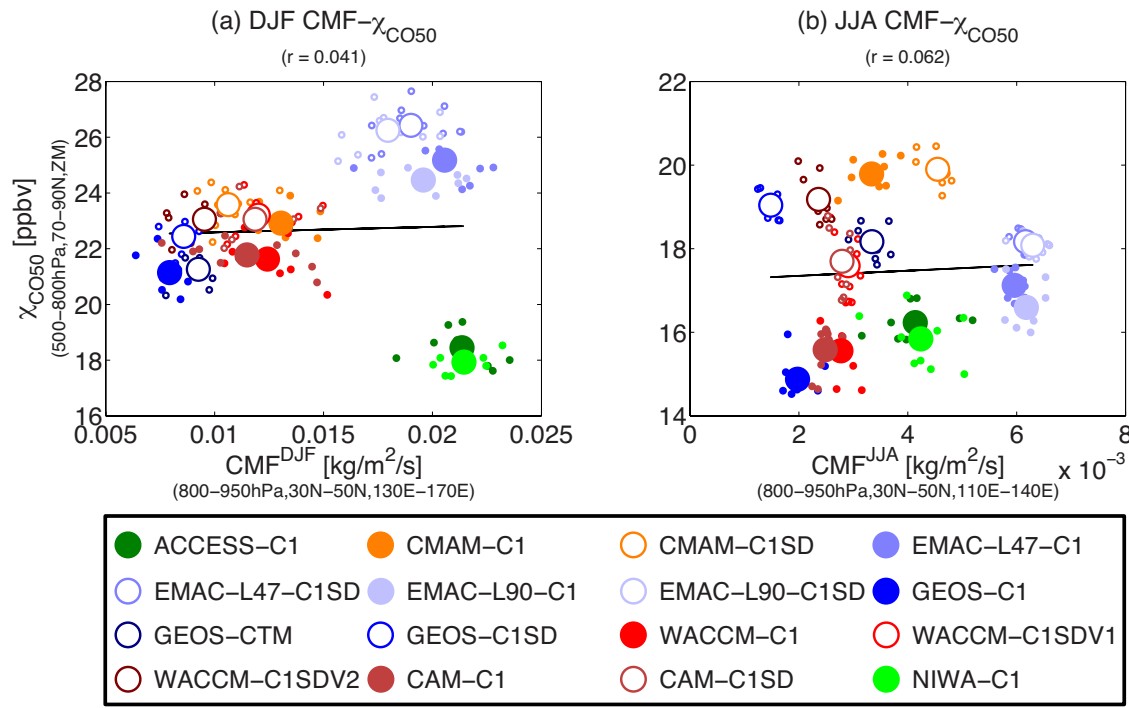

**Figure 3.** Correlation between Arctic CO50 concentration $\chi_{\text{CO50}}$ (units: ppbv) and low-level midlatitude CMF (units: kg/m$^2$/s) in (a) DJF and (b) JJA. Arctic $\chi_{\text{CO50}}$ is here the vertical average of 500 hPa - 800 hPa, latitudinal average of 70°N - 90°N, and zonal mean (ZM). CMF is here the vertical average of 800 hPa - 950 hPa, latitudinal average of 30°N - 50°N in both DJF and JJA, while longitudinal average window differs between seasons, as DJF CMF highlights the robust convection over the western Pacific Ocean (130°E-170°E) whereas JJA CMF focuses on the maritime convection over the East Asia (110°E-140°E) following Orbe et al. (2018). Large marks denote the 2000-2009 climatology (except GEOS-C1 and GEOS-C1SD) while small marks denote the corresponding interannual variations of each simulation. Results of least squares fitting based on climatological means are shown as the solid black lines, and the corresponding Pearson correlation coefficients are given in parentheses in the titles.





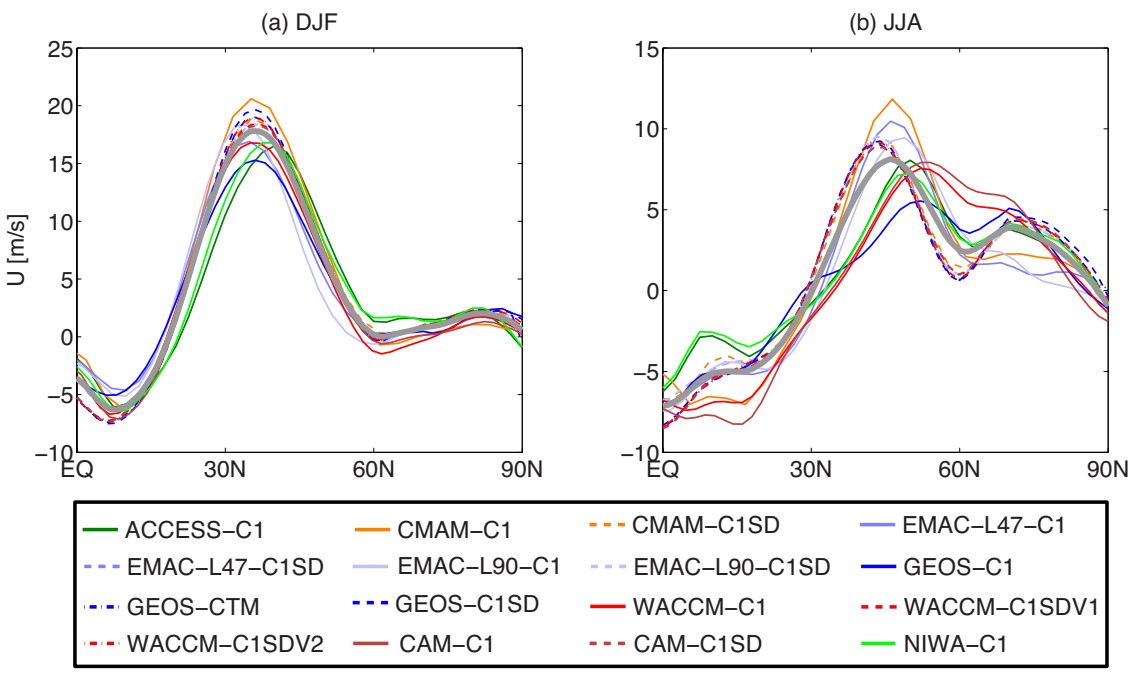

**Figure 4.** Multi-model spread of latitudinal profile of zonal wind $u$ vertically averaged between 500 hPa and 800 hPa and longitudinally averaged over the Pacific Ocean (135°E-135°W) during (a) DJF and (b) JJA. C1 simulations are shown in solid lines while C1SD simulations are shown in either dashed or dotted-dashed lines. The corresponding ensemble means are depicted as the heavy gray lines.





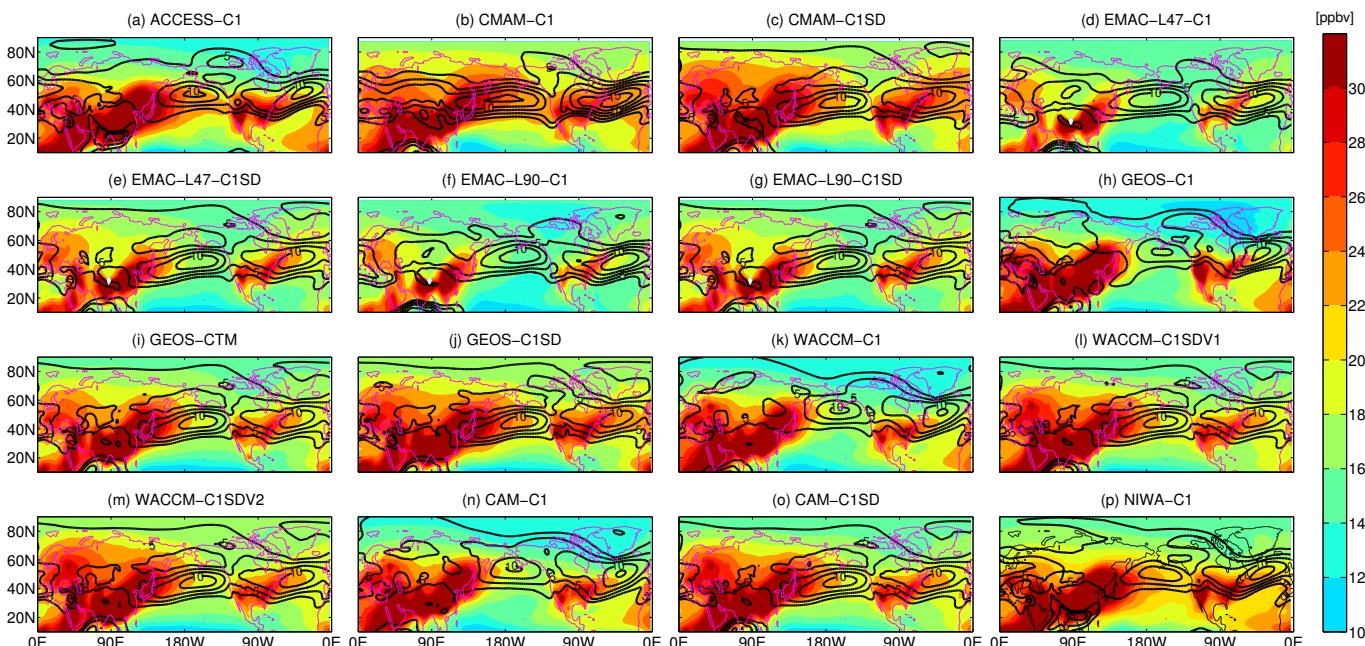

**Figure 5.** Maps of 500 hPa - 800 hPa averaged CO50 distribution (shades, units: ppbv) and the corresponding 500-800 hPa averaged $u$ during JJA in each CCMI simulation.





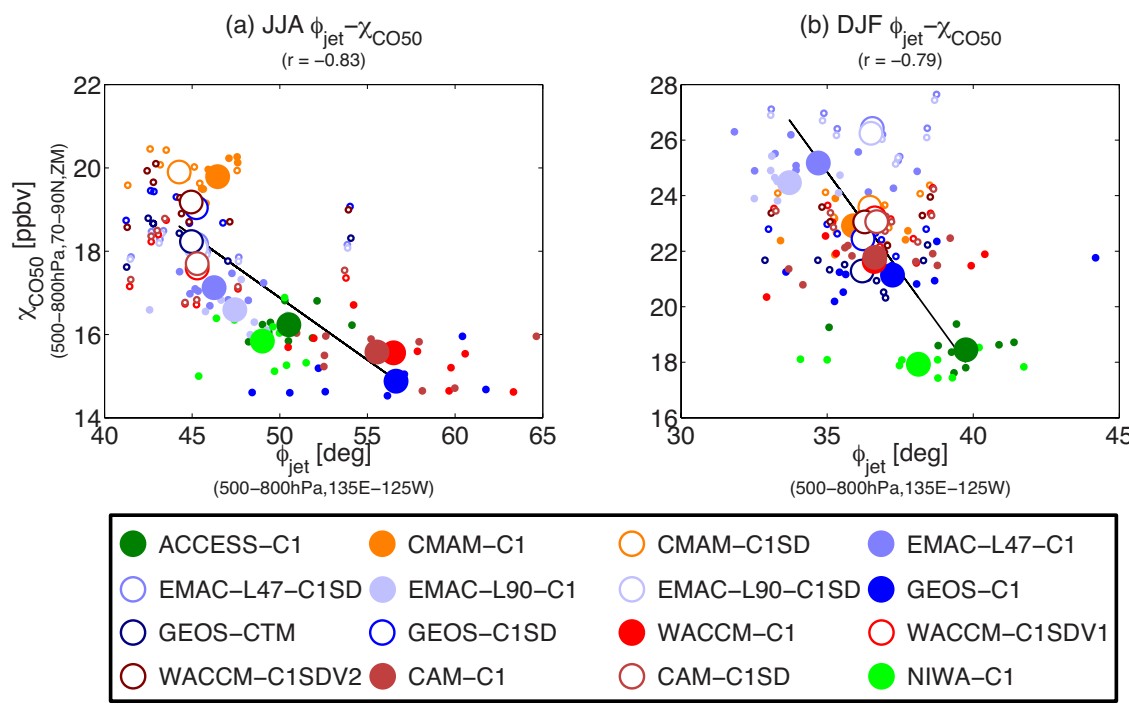

**Figure 6.** Similar to Fig. 3, but for the correlation between Arctic $\chi_{CO50}$ and latitudinal location of the NH midlatitude jet $\phi_{jet}$ over the Pacific Ocean (135°E-135°W) (Barnes and Polvani, 2013). Note that the sequence of displayed seasons switches with JJA in (a) and DJF in (b).





**Figure 7.** Schematics of mechanisms illustrating dynamic influences on the NH midlatitude-to-Arctic transport for the midlatitude jet situated more southern in (a) and more northern in (b). When jet location $\phi_{\text{jet}}$ and meridional flow switching point $\phi_{v=0}$ are more southern, cyclonic wave breaking (CWB) along the poleward flank of the jet and northward surface meridional flow (green arrow) result in high tracer concentrations in the high latitudes. In contrast, when $\phi_{\text{jet}}$ and $\phi_{v=0}$ are more northern, anticyclonic wave breaking (AWB) along the equatorward flank of the jet and southward surface meridional flow (blue arrow) result in low tracer concentrations over the Arctic. Tracer sources in the NH midlatitudes are denoted by the gray shades, isentropic surfaces are depicted as dashed lines, red and blue triangles mark the latitude of $\phi_{\text{jet}}$ and $\phi_{v=0}$ respectively.



**Figure 8.** Tracer flux diagnostics showing the total flux $\langle \overline{v \chi_{CO50}} \rangle$ (heavy), zonal mean flux $\langle \overline{v} \ \overline{\chi_{CO50}} \rangle$ (light, dashed) and eddy flux $\left\langle \overline{v' \chi'_{CO50}} \right\rangle$ (light, solid) of CO50 (vertically integrated from 1000 hPa to 200 hPa; positive means northward; units: kg/s) in (a) GEOS-C1, (b) GEOS-C1SD, and (c) the difference GEOS-C1SD − GEOS-C1 during summer. (d,e,f) are similar to (a,b,c) but during winter. Results are based on the daily GEOS model output.





**Figure 9.** Approximated zonal-mean flux using monthly output: (a,c) latitudinal profile of $\langle \overline{v}\ \overline{\chi_{CO50}} \rangle$ (units: kg/s); (b,d) correlation between the zonal-mean flux (latitudinal average of $40°$N - $60°$N) and the Arctic CO50 concentration (vertical average of 500 hPa - 800 hPa, latitudinal average of $70°$N - $90°$N, and zonal mean (ZM), units: ppbvl), during JJA in (a,b) and during DJF in (c,d).







**Figure 10.** Similar to Fig. 9 in (a,b,d,e), but for low-level (800 hPa - 950 hPa) zonal-mean meridional wind $v$ and $\phi_{v=0}$ marking the latitude for low-level $v$ switching from southward flow at south to northward flow at north. The correlations between $\phi_{v=0}$ and jet location $\phi_{jet}$ are further shown in (c,f).



**Figure 11.** As in Fig. 2 except for NH50.



**Figure 12.** (a,e) Tracer-tracer correlation between NH50 and CO50 over the Arctic (500 hPa - 800 hPa, 70°N-90°N, ZM), (b,f) as in Fig. 3, (c,g) as in Fig. 6, and (d,h) as in Fig. 10(b,e), except y-axis is replaced as the Arctic NH50 concentration. Results for JJA are shown in (a,b,c,d), while those for DJF are shown in (e,f,g,h)





**Figure 13.** Similar to Fig. 12(a,c,e,g), but for correlations: (a,c) between $\chi_{CO}$ and $\chi_{CO50}$, and (b,d) between $\phi_{v=0}$ and $\chi_{CO}$; during DJF in (a,b) and JJA in (c,d).