# Peer review of "Large-scale transport into the Arctic: the roles of the midlatitude jet and the Hadley Cell"

_Atmospheric Chemistry and Physics, 2018_

## Referee Comment (RC1) · Anonymous Referee #1 · 26 Sep 2018

This manuscript analyzed the transport of an idealized tracer CO50 from the northern midlatitudes to the Arctic among CCMI models. Large model spread in Arctic CO50 concentration is found among models with interactive dynamics as well as specified dynamics. The authors attributed this model spread to the location of low level divergence zone, which is also correlated with the location of Pacific jet. This study provides a useful summary of the CCMI model performance, and helps to advance our understanding of the tracer transport pathways into the Arctic, which is a key factor to understand the climate changes there. I find the mechanisms proposed by the authors to be reasonable, but I also find the evidences to support the mechanism to be somewhat weak and circumstantial. I understand that such situation may be unavoidable given the limited diagnostics from the multi-model simulations and the complex nature of the tracer

transport. Nevertheless, I listed my comments below for the authors to consider.

1. Most conclusions of this paper is based on correlation analysis of climatologies among models. As the authors already pointed out, these correlation are sensitive to the choices of models included into the analysis. Thus, even a seemly high and statistically significant correlation may not be robust after all. For example, the correlation between phi_v=0 and X_CO50 in JJA is 0.65 (Fig. 10 b). But if excluding CAM-C1, WACCM-C1 and GEOS-C1, the correlation becomes much weaker and may even changes sign.

2. The authors argued the mechanisms leading to the correlation between phi_v=0 and X_CO50 is that if the divergence zone locates north of the emission, then the northward transport from the source region is limited. This mechanism should work on the interannual time scales as well as the climatologies. Yet, the interannual variation within each model does not show similar correlations to the climatologies among different models. This is evident in Fig. 10e, which shows that the interannual variations of phi_v=0 within each model is comparable to the inter-model spread of climatology, but the corresponding Arctic CO concentration does not show any negative correlation. IN models such as CMAM-C1SD, the correlation from inter-annual variations even seems to be positive.

3. The authors argue the importance of the zonal mean circulation for the tracer transport by comparing the zonal mean and eddy component in two models: GEOS-C1 and GEOS-C1SD. But as shown in Fig. 9a, GEOS-C1 is clearly an outlier in terms of the zonal mean contribution to the tracer flux. If comparing GEOS-C1SD versus WACCM-C1 instead of GEOS-C1, the difference in the zonal mean flux would be much smaller. But the Arctic CO concentration is similar in WACCM-C1 and GEOS-C1, this would implies that the eddy component may be more important to explains the difference between the two models.

4. Since the argument is about the low level divergence zone (800-950 hPa), why the

tracer flux is integrated over the whole troposphere (200-1000 hPa) rather than just the low level? The mean circulation pointing to opposite direction in the lower and upper troposphere, and hence there may be canceling effects when integrating over the whole troposphere.

Editorial comments: P4L6: "blue contours" are missing in Fig. 1 P7L22: "SE-NW" I think the jet is tilting SW-NE, especially over western Pacific. P11 L16: This sentence sounds like strong correlation between NH50 and CMF is found in both winter and summer, with stronger value in winter. But in fact, it is only found in winter. There is no correlation in summer. P27 Fig. 8: the caption for dashed and solid lines are different from the labeling in the figure. Fig. 9 and 10: There is no need to include ACCESS and NIWA in the figure legend, since they are not included in these two figures.

---

## Referee Comment (RC2) · Anonymous Referee #2 · 11 Oct 2018

In this paper, the authors analyze three different tracers from the CCMI models in order to understand transport of tracers that are emitted in the NH to the Arctic. They find that transport from NH midlatitudes depends on whether sources are ocean or land-based and, in the case of land-based sources, on the climatology of the Pacific jet. They explain that this is because the jet is associated with the Hadley cell edge and associated meridional flow. Generally, I find this paper to be well written and clear. The result that the realistic CO and CO50 are not closely related is an important one, I think. It emphasizes the need to analyze the output of real chemical tracers and compare to observed tracer data. The differences between NH50 and CO50 are also important.

While the physical mechanisms proposed by the authors are plausible, the supporting evidence is based on correlations of decadal averages within a very limited number of

models rather than a clear causal indication of the process. It is clear that the number of models that can be examined is not the fault of the investigators, but given these constraints, it is not clear that any strong conclusions can be reached using these methods. In particular, the number of independent pieces of information is unclear. WACCM and CAM-1 seem nearly identical in the scatter plots. The EMAC-47 level model and EMAC-90 level model are definitely not independent, and considering the frequent grouping of these with their SD runs in the scatter plots, it would be useful to know how many independent data points these four represent. Since, as the authors note, the correlations can be easily influenced easily by which models are included the choice of weighting the single ACCESS run as equivalent to one of the EMAC runs, for example, could change the results significantly. I would expect that the mechanisms the authors point to would work on shorter timescales than decadal, so it is strange that the small dots of the annual averages do not fall closer to the relationships that are calculated. Perhaps an annual average is too short for the process to be valid, but why should a decadal average be necessary? Some filtered time series showing that this process is valid within a model and not just between models would be helpful, as it would help convince the reader that this physical mechanism is correct. Maybe GEOS has a longer record that could be used? And if the relationship only holds between models (i.e. the explanation of the jet and the tracer concentration only explains the difference between models, not the physical processes within the models), then it just implies that biases in models between transport of tracers and large scale circulation near the tracer sources are correlated.

The statistical methodology is problematic. The correlations are done using least squares, which necessarily holds one variable as dependent and one as independent. This is why all of the "fits" where there is no correlation are horizontal lines. Since there is no fundamental reason to expect that either quantity should be an independent variable, this method is not sufficient. Either the fit needs to be done with the x and y reversed (in y=ax+b) and that slope should be plotted as well, or a reduced major axis regression, with each quantity scaled by its variance, should be employed. With small

r values (e.g. Figure 3), no fit line should be shown at all.

The analysis of the differences between GEOS and GEOS SD is fine but since GEOS is shown to have quite different behavior from the other models in Figure 9, the conclusions drawn from this comparison are not obviously going to apply to other models.

My final concern is that analyzing transport in SD runs seems inherently problematic, since it is unclear whether they are actually conserving tracers. I assume that, since the authors are analyzing these, they have reason to believe that it is not a problem. A discussion of the extent to which the SD runs do or do not conserve mass and tracer concentrations would be helpful—I'm not sure if such a study already exists or if the authors could do some analysis of their own with the GEOS-SD model.

I wish I could provide more constructive feedback. Process-based analysis of the tracer transport in individual models seems necessary to demonstrate that the plausible mechanisms shown by the weak correlations are in fact the causal mechanisms. It seems like the authors could do this with GEOS, at the very least, and they could potentially get output from at least one other model to do the same analysis to demonstrate consistency. If the authors can address these substantial issues, I would be willing to review a revised manuscript.

---

## Author Comment (AC1) · 7 Dec 2018

**Reviewer 1**

We thank the reviewer for their comments. We agree with the reviewer that the evidence to support some of the conclusions regarding the mechanism is not as strong as we would like. However, as the reviewer commented, this is unavoidable given the limitation of a multi-model analysis. We have softened some of our conclusions and included more discussion/analysis of the robustness of our results. Responses to the reviewer's explicit comments are listed below (in red).

1. Most conclusions of this paper is based on correlation analysis of climatologies among models. As the authors already pointed out, these correlation are sensitive to the choices of models included into the analysis. Thus, even a seemly high and statistically significant correlation may not be robust after all. For example, the correlation between phi_v=0 and X_CO50 in JJA is 0.65 (Fig. 10b). But if excluding CAM-C1, WACCM-C1 and GEOS-C1, the correlation becomes much weaker and may even changes sign.

   We agree with the reviewer that the correlations are somewhat sensitive to the choice of the models. We had acknowledged this in the original version, but have made this clearer in the revised version. However, we think the correlations are actually fairly robust. We have calculated the correlation coefficients for different subsets of models, e.g., excluding SD, excluding models with similar heritage, and there are generally only small changes in correlations and often a higher correlation than using all models (e.g. using just the free-running models results in higher correlations between $\phi_{v=0}$ and CO-50, see Table R1). Excluding CAM-C1, WACCM-C1 and GEOS-C1 does have a big impact but we do not think this is a reasonable thing to do as these simulations are not physical outliers and their exclusion greatly reduces the range of $\phi_{v=0}$ and leaves only 2 independent FR models, so that resulting correlation is determined only by SD models. As we now make clear in the manuscript, the relationships are much less robust if only SD models are considered.

2. The authors argued the mechanisms leading to the correlation between phi_v=0 and X_CO50 is that if the divergence zone locates north of the emission, then the northward transport from the source region is limited. This mechanism should work on the interannual time scales as well as the climatologies. Yet, the interannual variation within each model does not show similar correlations to the climatologies among different models. This is evident in Fig. 10e, which shows that the interannual variations of phi_v=0 within each model is comparable to the inter-model spread of climatology, but the corresponding Arctic CO concentration does not show any negative correlation. In models such as CMAM-C1SD, the correlation from inter-annual variations even seems to be positive.

   We agree with the reviewer that the interannual variation between different metrics ($\phi_{jet}$, mean flux, and $\phi_{v=0}$) and CO50 for individual models generally doesn't show the same relationship as between models. This is now discussed in detail in the revised manuscript. There are two main reasons for this disagreement: (i) multiple processes influence the transport of CO50 into the Arctic, and although the mid-latitude mean meridional flux may dominate the variations between the models, other processes can have a larger influence on the interannual variation. (ii) The metrics used to define jet location and Hadley Cell edge can have large uncertainties for monthly or even seasonal data (e.g., jet can be weak with no distinct peak or v can be close to 0 over a broad latitude range, for given year) and differences in the metrics between two years may not be representative of the differences in the jet and HC width. Averaging over multiple years is required to get robust values of these metrics.

   An example of another process that influences the transport of CO50 into the Arctic is the meridional transport at high latitudes. The climatological meridional velocity and mean flux poleward of 70N are similar among models as compared to the intermodal variations in the midlatitudes (see Figures 9 and 10) and hence the transport difference due to advection by this flow does not contribute to the spread among models (see Table R2). However, there can be large interannual variations in the high latitude meridional flow/flux within individual models (see Figure R1). This plays an important role in the interannual variations in Arctic CO50 in most of the models with high or moderate interannual correlations between

high-latitude meridional flux and CO50 (see Table R2). This is one example, and there are likely other processes that make a larger contribution to interannual variations than to the spread among models.

As indicated above, another reason for the differences between interannual correlations and correlations among model climatologies is the large uncertainty / non-representativeness of metrics for individual seasons. A clear example of this is the $\phi_{v=0}$ during JJA. In many models, v is close to 0 for a wide range of latitudes in this season, and although there are only small differences in the mean meridional flow (and mean meridional flux) over this region between the years there can be large variations in $\phi_{v=0}$, see Figure R1. In particular, in the SD models the $\phi_{v=0}$ can vary between 30°N and 40°N for individual years, even though for all years there is a very weak meridional flow over this range of latitudes. As a result, the interannual spread in $\phi_{v=0}$ is not representative of interannual differences in meridional transport of CO50 between the years, and the interannual variations in $\phi_{v=0}$ (shown in Figure 10b) are generally larger than those in mean fluxes (Figure 9b).

3.  The authors argue the importance of the zonal mean circulation for the tracer transport by comparing the zonal mean and eddy component in two models: GEOS-C1 and GEOS-C1SD. But as shown in Fig. 9a, GEOS-C1 is clearly an outlier in terms of the zonal mean contribution to the tracer flux. If comparing GEOS-C1SD versus WACCM- C1 instead of GEOS-C1, the difference in the zonal mean flux would be much smaller. But the Arctic CO concentration is similar in WACCM-C1 and GEOS-C1, this would implies that the eddy component may be more important to explains the difference between the two models.

    We have followed the reviewer's suggestion below (comment 4) and now show in Figure 9 the vertically integrated mean flux calculates over the lowest levels (700-1000 hPa). GEOS-C1 is no longer an outlier, and the zonal-mean transport in GEOS-C1 is not structurally different from other models (e.g., GEOS-C1SD). Fig. 8 is also adjusted accordingly.

4.  Since the argument is about the low level divergence zone (800-950 hPa), why the tracer flux is integrated over the whole troposphere (200-1000 hPa) rather than just the low level? The mean circulation pointing to opposite direction in the lower and upper troposphere, and hence there may be canceling effects when integrating over the whole troposphere.

    This is a good point, and we have modified the figure as the reviewer suggested. There are no major changes to the conclusions, but the new figure should help ease the reviewer's concern on GEOS-C1 being as an outlier.

5.  Other Editorial comments

    We have revised the manuscript following reviewer's suggestions.

Reviewer 2

We thank the reviewer for their comments. We agree with the reviewer that the evidence to support some of the conclusions regarding the mechanism is not as strong as we would like, due to the limited number of models, and we have softened some of our conclusions and included more discussion/analysis of the robustness of our results. Responses to the reviewer's explicit comments are below (in red).

1. While the physical mechanisms proposed by the authors are plausible, the supporting evidence is based on correlations of decadal averages within a very limited number of models rather than a clear causal indication of the process. It is clear that the number of models that can be examined conclusions can be reached using these methods. In particular, the number of independent pieces of information is unclear. WACCM and CAM-1 seem nearly identical in the scatter plots. The EMAC-47 level model and EMAC-90 level model are definitely not independent, and considering the frequent grouping of these with their SD runs in the scatter plots, it would be useful to know how many independent data points these four represent. Since, as the authors note, the correlations can be easily influenced easily by which models are included the choice of weighting the single ACCESS run as equivalent to one of the EMAC runs, for example, could change the results significantly.

   We agree that it is not possible to make strong conclusions given the limited number of models, and we have softened our language so we are saying the analysis is suggestive rather than proves or shows. We also agree that the models are not all independent, and we have included discussion of the relations between models in the section 4.1. Specifically, we have now noted in the text the similarity between WACCM and CAM, between ACCESS and NIWA, and different EMAC versions. We have also calculated the correlation coefficients only using one of each of these pairs and the values are essentially the same to those using all the models (see Table R1).

2. I would expect that the mechanisms the authors point to would work on shorter timescales than decadal, so it is strange that the small dots of the annual averages do not fall closer to the relationships that are calculated. Perhaps an annual average is too short for the process to be valid, but why should a decadal average be necessary? Some filtered time series showing that this process is valid within a model and not just between models would be helpful, as it would help convince the reader that this physical mechanism is correct. Maybe GEOS has a longer record that could be used? And if the relationship only holds between models (i.e. the explanation of the jet and the tracer concentration only explains the difference between models, not the physical processes within the models), then it just implies that biases in models between transport of tracers and large scale circulation near the tracer sources are correlated.

   We agree with the reviewer that the interannual variation between different metrics ($\phi_{jet}$, mean flux, $\phi_{v=0}$) and CO50 for individual models generally doesn't show the same relationship as between models. Please see our response to comment #2 from reviewer 1, who raised the same issue.

3. The statistical methodology is problematic. The correlations are done using least squares, which necessarily holds one variable as dependent and one as independent. This is why all of the "fits" where there is no correlation are horizontal lines. Since there is no fundamental reason to expect that either quantity should be an independent variable, this method is not sufficient. Either the fit needs to be done with the x and y reversed (in y=ax+b) and that slope should be plotted as well, or a reduced major axis regression, with each quantity scaled by its variance, should be employed. With small r values (e.g., Figure 3), no fit line should be shown at all.

   We followed the reviewer's suggestion and used a total least square method (Petráš and Bednárová, 2010) to calculate the linear fit. Note that the Pearson's correlation coefficient is not affected by method of

fitting. We have also followed the reviewer's suggestion and do not show the linear fit when the correlation is not significant.

4. The analysis of the differences between GEOS and GEOS SD is fine but since GEOS is shown to have quite different behavior from the other models in Figure 9, the conclusions drawn from this comparison are not obviously going to apply to other models.

We have followed comment #4 by reviewer 1 and now show in Figure 9 the vertically integrating the mean flux only over low levels (700-1000 hPa), which is the most relevant for transport away from the source. In the new Figure 9 GOES-C1 is no longer an outlier, and therefore the proposed mechanism of location shift in mean meridional circulations and associated changes in low-level meridional flow is still applicable to the difference between GEOS-C1 and GEOS-C1SD.

5. My final concern is that analyzing transport in SD runs seems inherently problematic, since it is unclear whether they are actually conserving tracers. I assume that, since the authors are analyzing these, they have reason to believe that it is not a problem. A discussion of the extent to which the SD runs do or do not conserve mass and tracer concentrations would be helpful. I'm not sure if such a study already exists or if the authors could do some analysis of their own with the GEOS-SD model.

Mass conservation is not inherently problematic for the SD runs. These simulations used the same numerical framework as the free-running (FR) counterpart, so have the same transport algorithm and same implementation of emissions and chemistry as the FR. The only difference between a pair of SD and FR simulations is the winds that are used in the transport. While there may be conservation issues with the winds directly from the reanalyses, in all but one of the SD runs the horizontal winds are only nudged towards the reanalyses (and not set directly equal to the reanalyses) and winds are constrained together with the vertical velocity to conserve mass. The one exception is the GEOS-CTM, the only CTM analyzed in this study, which uses horizontal and vertical winds directly from the reanalyses, and adjusts surface pressure to conserve mass. To reassure the reviewer, we calculated the global $CO50$ burden in each CCMI simulation (see Table R3), and only small differences can be found among models (except GEOS-CTM), with these small differences likely due to due to interpolation of the emissions onto different resolutions of model grids. In particular, when considering the C1-C1SD pairs within the same base model, the differences are very small (less than 1%).

[Figure]

Figure R1 Latitudinal profile of JJA low-level zonal mean meridional wind $v$ $(800-950$ hPa) in each simulation. The black lines and circles denote the interannual variations of $v$ and $\phi_{v=0}$ within the model, while gray shades give the multi-model spread among model climatology, as shown in Figure 10(a). The blue thick line denotes the climatological $v$ of the simulation.

Table R1 Correlation coefficients between Arctic CO50 concentration and physical process related metrics (such as $\phi_{jet}$, mean flux, and $\phi_{v=0}$ as listed in the 1st column) calculated using all available models versus only the free-running (C1) models versus clustered models (i.e., exclude NIWA-C1 for similarity to ACCESS-C1, exclude EMAC-L90-C1/C1SD for similarity to EMAC-L47-C1/C1SD, exclude WACCM-C1SDV2 for similarity to WACCM-C1SDV1, and exclude CAM-C1/C1SD for similarity to WACCM-C1/C1SDV1). Calculation of correlation is based on climatology, and those are statistically significant (95%) are marked in **bold**.

| | DJF | | | JJA | | |
|---|---|---|---|---|---|---|
| | All | C1 | Cluster | All | C1 | Cluster |
| $\phi_{jet}$ | **-0.63** | **-0.92** | **-0.64** | **-0.84** | **-0.79** | **-0.83** |
| mean flux | **0.69** | 0.68 | **0.69** | **0.78** | **0.96** | **0.79** |
| $\phi_{v=0}$ | **-0.76** | **-0.95** | **-0.76** | **-0.58** | **-0.70** | **-0.51** |

Table R2 Correlation coefficients between mean meridional flux over high latitudes (60°N-80°N) and Arctic CO50 concentrations (500-800 hPa, 70°N-90°N, zonal mean). For mean meridional flux over high latitudes, it is vertically integrated in the low levels (700-1000 hPa) during DJF but in the upper levels (300-500 hPa) during JJA, considering differences in the CO50 vertical maximum between seasons (Fig. 1 and Fig. 2(c,d)). The correlation among model climatologies are shown in the 2nd row (similar to results shown in Fig. 9 (b,d)), while interannual correlations in individual simulations are shown in the 3rd – 16th rows. Coefficients that are statistically significant (95%) are marked in **bold**.

| | DJF | JJA |
|---|---|---|
| Multi-model Climatology | 0.28 | 0.37 |
| | | |
| CMAM-C1 | **0.67** | -0.26 |
| CMAM-C1SD | 0.35 | **0.74** |
| EMAC-L47-C1 | 0.50 | 0.22 |
| EMAC-L47-C1SD | 0.43 | 0.60 |
| EMAC-L90-C1 | 0.51 | -0.25 |
| EMAc-L90-C1SD | 0.43 | 0.55 |
| GEOS-C1 | **0.88** | **0.70** |
| GEOS-C1SD | 0.12 | **0.75** |
| GEOS-CTM | 0.46 | **0.84** |
| WACCM-C1 | -0.02 | 0.52 |
| WACCM-C1SDV1 | 0.09 | **0.78** |
| WACCM-C1SDV2 | 0.53 | **0.83** |
| CAM-C1 | 0.50 | 0.51 |
| CAM-C1SD | 0.15 | **0.75** |

Table R3 Climatological global CO50 burden (units: $\times 10^{10}$ kg) in each CCMI simulation. C1-C1SD pairs in the same base model are highlighted in the same box. Note that molecular weight of CO50 is assumed to be the same as CO (28g/mol).

| Model | Global CO50 burden ($\times 10^{10}$ kg) |
|---|---|
| ACCESS-C1 | 6.18 |
| CMAM-C1 | 6.39 |
| CMAM-C1SD | 6.40 |
| EMAC-L47-C1 | 6.35 |
| EMAC-L47-C1SD | 6.37 |
| EMAC-L90-C1 | 6.38 |
| EMAC-L90-C1SD | 6.40 |
| GEOS-C1 | 6.42 |
| GEOS-C1SD | 6.46 |
| GEOS-CTM | 5.95 |
| WACCM-C1 | 6.22 |
| WACCM-C1SDV1 | 6.21 |
| WACCM-C1SDV2 | 6.20 |
| CAM-C1 | 6.24 |
| CAM-C1SD | 6.23 |
| NIWA-C1 | 6.18 |

---

## Author Response (AR2)

We thank both reviewers' comments on the manuscript. We have listed our point-by-point responses below with black text showing the reviewers' comments and red text showing our responses.

Reviewer #1

This manuscript discusses the simulated tracer transport from midlatitude to the Arctic in CCMI models. The authors showed that for an idealized tracer CO50, large inter-model spread is found in its concentration over Arctic with or without specified dynamics. Unlike tracer NH5 reported in earlier studies, this inter-model difference cannot be explained by parameterized convection, but is found to be correlated with midlatitude jet position and Hadley cell boundary. Long-range transport of chemically and radiatively important species are important issues that lacks of full understanding. This manuscript provides a useful assessment of the CCMI models' performance on this subject. In this revision, the authors have made some efforts to improve the robustness of their results. I have some remaining minor comments for the authors to consider.

The "20%-40%" inter-model spread of the Arctic CO50 concentration. This number is quoted in the abstract and the conclusion, but it is not very clear where this number comes from. From the paragraph on page 6 line 4-13, it seems this number is calculated as the range (max-min) of the Arctic mean CO50 concentration divided by its multi-model mean. In this paragraph, the range is quoted to be ~11ppbv in DJF. Estimated from Fig. 2, the multi-model mean Arctic CO50 concentration is between 20 to 25 ppmv below 400 hPa. Then the relative inter-model spread would be 44%-55%. Is it because the authors define the inter-model spread as something else than the range? Perhaps the standard deviation among models? In any case, the authors need to articulate how they calculate this inter-model spread.

We define the inter-model spread as the range (max-min) and the numbers of fraction are given as the ratio of such a multi-model range to the multi-model mean. Here, we re-examined the range, mean and fractional ratio for Arctic CO50 concentrations in Table R1. The fractional ratio is about 30%-45% in DJF and 25%-30% in JJA. Therefore, we have made it more clearer on how these fractions are defined and we also revised the associated numbers slightly in the manuscript.

Table R1 Multi-model range, mean, and fraction ratio (range/mean) of the Arctic CO50 concentration (70N-90N).

| | DJF | | | JJA | | |
|---|---|---|---|---|---|---|
| | Range [ppbv] | Mean [ppbv] | Ratio | Range [ppbv] | Mean [ppbv] | Ratio |
| 500 hPa | 9.73 | 21.72 | 0.45 | 4.92 | 18.81 | 0.26 |
| 550 hPa | 9.39 | 21.95 | 0.43 | 4.96 | 18.43 | 0.27 |
| 600 hPa | 9.08 | 22.13 | 0.41 | 5.14 | 18.03 | 0.29 |
| 650 hPa | 8.69 | 22.35 | 0.39 | 5.21 | 17.58 | 0.30 |
| 700 hPa | 8.20 | 22.65 | 0.36 | 5.20 | 17.10 | 0.30 |
| 750 hPa | 7.58 | 23.04 | 0.33 | 5.12 | 16.60 | 0.31 |
| 800 hPa | 6.87 | 23.59 | 0.29 | 4.95 | 16.02 | 0.31 |
| 500-800 hPa Average | 8.51 | 22.49 | 0.38 | 5.07 | 17.51 | 0.29 |

Throughout the manuscript, the authors are doing averages over difference latitudes and pressure levels. It is not always clear why the authors made these choices. For example, CO50 concentration is averaged over 500-800 hPa, but the tracer flux is averaged over 700-1000 hPa, and the convective mass flux and the low level meridional winds are averaged over 800-950 hPa. The tracer mass flux is averaged over 40°N-60°N for JJA but 30°N-50°N for DJF. The authors need to justify their choices.

Thanks for the suggestion, and we have added some justifications throughout the revised manuscript. After the revision, we only apply two different pressure windows for vertical average. One is 500-800 hPa for examining the tracer concentration and zonal wind in the middle and lower troposphere. The middle and lower troposphere is the critical layer for trace gases to exert direct radiative impacts as well as indirect radiative impacts via interaction with clouds, and therefore it is of great interest to examine their concentrations at these levels. For zonal wind, middle and lower troposphere is the layer where the midlatitude jet streams can have a significant impact on tracer transport via wave breaking. The other vertical average zone is 800-950 hPa for meridional wind, flux, and convective mass flux, as all these metrics are found important at low levels. As to the difference of meridional window for tracer mass flux between JJA and DJF, this is because, on average, the maximum poleward zonal-mean flux occurs about 10° further north during JJA than that during DJF (see Fig. 9), and it is more reasonable to measure the tracer flux over regions where a similar distance from the Hadley Cell is kept.

Page 5 Line 27: I don't see how Fig. S1 support the argument in this sentence. Fig. S1 does not show the emission nor convection.

We have made a new Fig. S1 to show relations among CO50, emission, and convection. One important note for interpreting the secondary maximum of CO50 concentration in the subtropics (~30N) during summer is the difference between regional and zonal average. Over the convective plume where CO50 emission and convection are generally collocated, CO50 concentration still decreases as altitudes increase. However, in the perspective of zonal mean as shown in Fig.1 as well as Fig.S1(b) (also as Fig. R1(b)), zonal-mean CO50 concentration is lower at lower levels due to large compensation by low CO50 concentration outside the convection plume, and the zonal-mean concentration is higher at higher levels due to smaller zonal tracer gradient. Although interesting, this argument on the "secondary maximum" is a side note and we decide to keep the statement in a concise format with the newly made Fig. S1 for a better illustration.

[Figure]

Fig. R1 Relations among emission, convection, and CO50 concentration during JJA: (a) longitudinal distribution of CO50 at 30N for various vertical levels from 200 hPa to 500 hPa; (b) vertical profile of zonal-mean CO50 concentration at 30N highlighting the values at a few vertical levels that are shown in (a); and (c) 500 hPa CMF (black) and surface CO50 emissions (gray) at 30N.

Page 5 Line 5-6: From Fig. 2, it is not obvious that the inter-model range in JJA peaks around 400 hPa. It seems the JJA inter-model range remains relative constant at all levels below 400 hPa.
Yes, we have revised the text as suggested.

Page 8 Line 4: why the word "strong" needs to be quoted?
We removed the quote.

Page 21 Table 2 caption: Why some variables are scaled? For the SD simulations without a superscript note, are they using a common nudging time scale? What is the difference between the "replay" simulation and the standard nudging simulation?

Some variables in Table 2 are scaled because a few models have not comply with the standard boundary condition of NH50 mixing ratio as 100 ppbv. For example, the boundary condition is set as 100 ppmv in the GEOS-C1/C1SD and therefore needs to be scaled by 0.001. Similarly, CMF output in ACCESS-C1 and NIWA-C1 is multiplied by the gravitational coefficient g ~ 9.8 [m/s$^2$] and thus a scaling is applied.

Nudging time scales in SD simulations actually vary from different modeling centers, and not as a common time scale (say 50 hrs). We only highlight the two in the WACCM C1SD because WACCM is the only model that implement two different nudging time scales. We decided to remove these footnotes on nudging time scales and instead state them in the caption.

The "replay" simulation involves reading in analyzed field every 6 hrs and recomputing the analysis increment using the same assimilation methods that produce the MERRA reanalysis. Details on the "replay" approach are referred to Orbe et al. (2017).

Orbe, C., Oman, L. D., Strahan, S. E., Waugh, D. W., Pawson, S., Takacs, L. L., & Molod, A. M. (2017). Large-scale atmospheric transport in GEOS replay simulations. Journal of Advances in Modeling Earth Systems, 9. https://doi. org/10.1002/2017MS001053

Page 22 Fig. 1 caption: panel (e) and (f) do not show the "meridional and vertical transport of CO50". They only show the zonal mean CO50 concentration cross sections.

Revised as suggested.

Reviewer #3

This is an interesting paper aimed at understanding the cross-model variation in the transport in to the Arctic of tracers with mid-latitude sources. Following several other works that have pointed out the diversity in Arctic abundances of these tracers, this work points out convincingly the importance of transport by the zonal mean, meridional circulation for understanding this diversity in the CCMI multi-model ensemble. An important wrinkle to this is that this meridional transport differs even across models that specify the meteorology in some way. Although the role of the mean meridional circulation does not explain all aspects of inter-model disagreement regarding large-scale transport into the Arctic, it seems to be a significant contributing factor, and well worth highlighting.

I find this to be an interesting study well worth publishing. I have some questions about the methodology and some suggestions about presentation, but if these are addressed I would recommend publication. I also found the writing to be a bit difficult to follow at times.

My only more significant comment is regarding the choice of flux decomposition into a zonal mean an eddy component. I was initially somewhat surprised at this choice, rather than decomposing into a stationary (say,monthly mean) and time-varying component, as it seems the latter would be better suited for understanding the local poleward transport in the Pacific sector. The time mean would filter out transport from synoptic scale eddies just as well. I would have thought the zonal mean meridional velocity would be just as, if not more, relevant to NH50 transport as to CO50, but that doesn't seem to be the case (I appreciate the CMF argument and this may well be the story). Perhaps the zonal decomposition is easier to study with the model output available (though this isn't so clear to me). Have you tried this decomposition?

The aim of the decomposition into a zonal mean and an eddy component is to differentiate the roles of mean meridional circulation and atmospheric waves respectively in the poleward transport of tracers. Given that the focus shifts to the zonal-mean component later in the paper, it is also easier to implement the analysis to a larger group of CCMI models with only monthly-mean output. Instead, for a temporal decomposition into a stationary and a transient component, this would be more applied to just the eddies since the zonal-mean tracer flux is mostly contributed by the stationary component, and also the analysis can only be implemented in the GEOS simulations in which daily output are available. As we have shown in Fig. 8, for the total tracer flux difference between GEOS-C1 and GEOS-C1SD, the contribution by difference in the eddy component (stationary + transient) is minor. Therefore, by implementing the temporal decomposition, even if a contribution by synoptic eddies is found, that contribution must be largely cancelled by stationary eddies. This is confirmed in Fig. R2 that we show four components of both temporal and zonal decomposition (compare red dashed for stationary eddies and blue dashed for transient eddies). Note that we have modified the vertical integral range of tracer flux to $800 - 950$ hPa (to be consistent with the range where we examine $v$ and CMF) in the revised paper.

[Figure]

Fig. R2 Similar to Fig.8, but showing all four components from both zonal and temporal decomposition. The four components are: stationary zonal-mean flux (red solid), stationary eddy flux (red dashed), transient zonal-mean flux (blue solid), and transient eddy flux (blue dashed).

Given that no substantial changes are made to conclusions in the paper if imposing an additional temporal decomposition, we decided to keep the discussions on the zonal decomposition.

On a related note, it seems that a second way to quantify the contribution of the meridional transport (beyond inter-model correlations) would be to directly calculate the tracer flux from the time mean (or zonal mean) meridional winds. I would expect this to have substantial variations even in CO and in NH50, even if there are confounding effects that reduce the correlations with overall Arctic burdens. This could be quite helpful for quantifying the importance of this effect for the transport of realistic tracers. Possibly one could even try isolating the role of inter-model differences in meridional velocity by computing the flux with a reference tracer distribution.

We appreciate the idea of directly calculating tracer flux and we have done so for the CO50 tracer (see Fig.9 b,d). As for NH50 and CO, we have not explicitly calculated the zonal-mean flux but we are highly confident that the tracer flux will deliver a similar story as diagnosed by the correlation between Arctic tracer concentration and the Hadley Cell northern edge ($\phi_{v=0}$). Given that NH50 and CO are not the primary foci of the paper, we decide to keep the current form of analysis.

With regards to the presentation, I wonder if it would help to better distinguish between model integrations with constrained meteorology and those with free running dynamics. I have included specific suggestions below.

Please see responses in the "specific comments".

Specific comments

p3 l29 - Are the source data for this interpolation really on the model native grid? I would be surprised if any model runs in isobaric coordinates rather than hybrid pressure (this seems to be confirmed by Table 1). I suspect it's more likely that the source data have been interpolated by the modeling centers.

Yes, we have modified the text to make this issue more clearer.

p6 l10-11: I was confused by the sentence starting 'However, it is difficult....' Is the point that these integrations are different because of how they implemented the source of CO50, but that they may also differ in their transport?

We apologize for the confusion. We mean that, in EMAC simulations, it is difficult to reduce the Arctic CO50 concentration uncertainty caused by biases in the CO50 emissions with a simple scaling due to complexity in the poleward transport. We have modified this sentence accordingly.

p6 l15: I can see how this source variability increases the model range in winter, but the EMAC models are right in the middle of the model distribution for summer Arctic burdens so the latter assertion wasn't so clear to me.

As discussed above, even inputting equal magnitude of biases (with opposite directions) in CO50 emissions between winter and summer, the responses in the Arctic CO50 concentration may still differ between seasons due to different patterns of long-range poleward transport. As illustrated in Fig.1, poleward transport of CO50 are more along isentropic surfaces during winter while the transport displays a robust convection-related feature during summer and transport at higher levels might also be important. Alternatively speaking, during winter, the midlatitude surface source and the polar middle and lower troposphere are more directly linked via slopes of isentropic surfaces while this connection via relatively fast isentropic mixing seems to be weaker during summer. Therefore, biases in the midlatitude source can be more clearly seen in the Arctic CO50 concentration during winter than summer. Since the issue discussed above is not closely relevant to the statements in P6L15, we decided to make no revisions.

p7 l20-24: I wasn't so convinced by this argument - even if the transport into the Arctic is weaker in the summer, if the sources are at midlatitudes, this transport still seems essential for determining summer burdens, no? Unless a large fraction of the summer is left from winter-time transport, in which case this is probably worth discussing, since transport in the winter and spring would be more important than summer time transport.

Yes, the statement here is mostly speculative. We only provide one plausible explanation on why we have not seen a robust correlation between Arctic CO50 concentration and midlatitude convection during summer even when there is a collocation between summertime convection and CO50 source over lands. It is that isentropic transport is much weaker during summer. Although CO50 concentration over regions slightly higher above the midlatitude source can be highly modulated by variations in the parameterized convection. This variation at the lower end of isentropic slopes can be less dominant in influencing the Arctic CO50 burden at the higher end of isentropic slopes with less efficient isentropic transport during summer. Other processes may also be influencing resulting in the worse correlation between midlatitude convection and Arctic CO50 concentration during summer seen in Fig. 3b. We think the current statement is clear enough so decide to make no revision. As to the reviewer's comments on leftover effect of

winter-time transport on summertime Arctic burden, we think the chance is smaller as the lifetime of the tracer is 50 days, which means that tracers are well chemically diluted at a seasonal scale. Therefore, it is reasonable to assume that variations in the Arctic tracer burden are by majority contributed from variations in the transport during the same season.

p9 l12: This is inconsistent with the caption of Fig. 8 that gives an upper bound is 200 hPa. Thanks for pointing out the typo. Note that the integral bounds have been changed to $800 - 950$ hPa in the revised paper.

p12 l15: By 'unchanged' do the authors mean that the model anomaly from the ensemble mean NH50 concentrations at high latitudes is consistent with its anomaly at midlatitudes? Yes.

For all figures I found myself wishing there was a clearer sorting in the legends by specified versus free dynamics. I don't think it is said anywhere explicitly that the open circles are specified dynamics integrations. Also, where correlation coefficients are given it would be informative to show coefficients for only the specified/free dynamics runs. Some of the regression lines (e.g. Fig. 10c) seemed to be obscuring clear differences in behavior; similarly in panels 12d,e,h, for instance, the two sets of integrations seem to follow rather different regression lines. This is discussed at some points in the text, but the presentation could be made clearer.
We appreciate the reviewer's suggestion on sorting the legends by C1 versus C1SD. We have not revised the legend but instead made statement in the captions when firstly noted to make it clearer the difference between lines and marks used to differentiate C1 and C1SD simulations. As to correlation coefficients, we prefer the current way for all models although sometimes such a way could be obscuring. As discussed in the revised paper, the correlation coefficients are only for referencing given that they are derived based on climatologies from a limited number of simulations. Also, we have indicated the difference in correlations based on all simulations versus that is only based on the C1 simulations in Table S1, and we have highlighted this point more throughout the revised manuscript.

Fig. 2,4, and similar panels in the following: Would it be helpful to show ensemble means separate for specified dynamics and free running models?
We aim to examine C1 and C1SD simulations together in these figures, and by showing the ensemble mean for all models, it is easier to derive the multi-model fractional spread (of CO50 concentration) noted in the paper. We tried to explicitly show the separation between C1 and C1SD simulations for the CO50 concentration in Fig. S2. Also, adding more lines may not help in distinguish different lines for different simulations.

Fig. 5: In this figure especially, it is hard to take in these 16 panels - sorting them by specified versus free dynamics would be more useful than by modeling center.
This is a brilliant idea. We have re-arranged the panels and make the eight C1 simulations at the top followed by the other eight C1SD simulations at the bottom.